# 🚐 SUV: Scalable Large Language Model Copyright Compliance with Regularized Selective Unlearning

**Tianyang Xu,**\* **Xiaoze Liu,**\* **Feijie Wu, Xiaoqian Wang, Jing Gao**
Purdue University
{xu1868, xiaoze, wu1977, joywang, jinggao}@purdue.edu

## Abstract

Large Language Models (LLMs) have transformed natural language processing by learning from massive datasets, yet this rapid progress has also drawn legal scrutiny, as the ability to unintentionally generate copyrighted content has already prompted several prominent lawsuits. In this work, we introduce SUV (Selective Unlearing for Verbatim data), a selective unlearning framework designed to prevent LLM from memorizing copyrighted content while preserving its overall utility. In detail, the proposed method constructs a dataset that captures instances of copyrighted infringement cases by the targeted LLM. With the dataset, we unlearn the content from the LLM by means of Direct Preference Optimization (DPO), which replaces the verbatim copyrighted content with plausible and coherent alternatives. Since DPO may hinder the LLM's performance in other unrelated tasks, we integrate gradient projection and Fisher information regularization to mitigate the degradation. We validate our approach using a large-scale dataset of 500 famous books (predominantly copyrighted works) and demonstrate that SUV significantly reduces verbatim memorization with negligible impact on the performance on unrelated tasks. Extensive experiments on both our dataset and public benchmarks confirm the scalability and efficacy of our approach, offering a promising solution for mitigating copyright risks in real-world LLM applications. Code is available at https://github.com/xz-liu/SUV/.

## 1 Introduction

Large Language Models (LLMs) have demonstrated impressive utility by generating human-like text from massive datasets. However, their widespread adoption has been accompanied by serious copyright concerns (Huang et al., 2025; Min et al., 2023; Henderson et al., 2023; Liu et al., 2024c; Chen et al., 2024), exemplified by high-profile lawsuits (Adams, 2023; Maheshwari & Tracy, 2023; Tracy & Maheshwari, 2023). Numerous studies have confirmed that LLMs can memorize and regurgitate verbatim excerpts of copyrighted materials (Chang et al., 2023; Karamolegkou et al., 2023; Liu et al., 2024c; Wei et al., 2024). In an ideal world, one would train LLMs solely on non-copyrighted data, yet the sheer scale and labor required for data curation render this approach impractical, and retraining models from scratch without copyrighted material is equally inefficient. Consequently, various post-training "patching" methods have been proposed to address this issue.

Existing approaches to preventing copyright infringement in LLM outputs can be broadly divided into three categories. First, *decoding-based methods* (Wei et al., 2024; Ippolito et al., 2023b; Abad et al., 2024b; Vyas et al., 2023) modify the next-token probability distribution during decoding. While these methods can reduce the risk of generating verbatim outputs, they can be easily removed by attackers since they are not integrated into the model itself. Furthermore, some of them may be prone to inducing hallucinations (Liu et al., 2024c; Abad et al., 2024b). Second, *agent-based methods* (Liu et al., 2024c; Hua et al., 2024) build external agents to detect and prevent the generation of illegal content. Although these

---

\* These two authors contributed equally; order was determined randomly (by rolling a die).

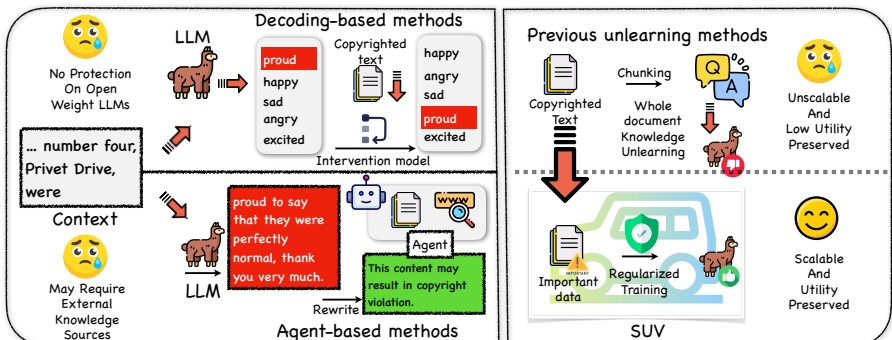

Figure 1: Comparison between different copyright compliance approaches for LLMs

methods accurately identify problematic outputs, they often require additional resources (e.g., internet access) and fail to remove the memorized content embedded within the model itself. Third, *unlearning methods* (Bourtoule et al., 2021; Maini et al., 2024; Dou et al., 2025; Zhang et al., 2024; Sekhari et al., 2021; Xu et al., 2023; Liu et al., 2024d) directly target the removal of memorized copyrighted content from the model, which have shown promise for open-weight LLMs; however, they face several challenges:

- *Utility degradation:* Many existing approaches achieve unlearning by erasing the knowledge of entire copyrighted material, aiming to remove not only copyrighted information but also information that can be generalized or rephrased from the original ones. However, existing copyright infringement cases are primarily related to verbatim excerpts (Karamolegkou et al., 2023; Liu et al., 2024c; Wei et al., 2024), indicating that existing approaches may be too strict and may overcorrect for many cases. Still, verbatim memorization unlearning is also likely to undermine LLM utility (especially the performance of unrelated tasks). With the increasing scale of copyrighted materials, existing unlearning approaches exhibit significant drops in utility, rendering them unscalable.
- *Dataset limitations and scalability:* Currently available benchmarks are generally small-scale and predominantly based on non-copyrighted materials. Researchers (Liu et al., 2024c) have shown that non-copyrighted materials tend to have higher memorization rates than copyrighted ones, meaning that using non-copyrighted datasets for alternative evaluation may not accurately reflect the evaluation of copyright violations due to differing data distributions. However, due to copyright restrictions, assembling large-scale datasets of copyrighted material is challenging.

To overcome these challenges, we propose SUV (Selective Unlearing for Verbatim data), a novel unlearning framework that adopts a *selective* approach, motivated by two key observations. First, the context generation for copyrighted materials is *highly localized*—unlike general domain knowledge, verbatim memorization is confined to specific passages, making broad unlearning both unnecessary and potentially harmful. Second, memorization follows a *long-tail distribution*, meaning that only a small subset of the training data is responsible for most copyright risks. These insights motivate a targeted unlearning strategy that focuses exclusively on the problematic segments and parameters.

Our approach unfolds in three primary stages. *Firstly*, unlike previous methods that segment text into large chunks, we employ a sliding-window mechanism to achieve fine-grained identification of memorized passages. This allows us to apply a more granular approach in distinguishing between relevant and irrelevant information within the model's memory. *Secondly*, we tackle the challenge of utility degradation. Many existing approaches remove not only copyrighted content but also information that can be generalized or rephrased from the original texts. Consequently, unlearning verbatim memorization can undermine LLM utility—especially on unrelated tasks—and lead to significant performance drops as the scale of copyrighted materials increases. In contrast, our method unlearns the targeted segments via Direct Preference Optimization (DPO) (Rafailov et al., 2024), preferring a randomly generated completion over plagiarized content. Although the random completion may initially yield meaningless passages, we mitigate the degradation by integrating projected gradient techniques and Fisher information regularization to control the unlearning process

and prevent dramatic LLM updates. In contrast to additive-loss baselines such as GA and NPO that simply *sum* separate "forget" and "retain" objectives, our DPO formulation turns unlearning into a *single, preference-based* optimisation problem. This unified objective updates only the parameters most responsible for verbatim memorisation, thereby minimising collateral damage to general knowledge and downstream utility. *Finally*, we address dataset limitations and scalability by constructing a large-scale dataset comprising 500 famous books—predominantly copyrighted works such as *Harry Potter Series* and *A Song of Ice and Fire Series*—to faithfully mimic real-world usage. By assembling our dataset of 500 copyrighted books, our approach provides a more realistic assessment of LLM behavior under copyright constraints.

By addressing the limitations of existing approaches, our work provides a scalable and practical solution for mitigating copyright infringement in LLMs while ensuring that their general-purpose capabilities remain largely intact. *Our contributions are as follows:*

- To the best of our knowledge, this is the largest-scale study on unlearning copyrighted texts, leveraging a dataset of 500 well-known books. Our dataset comprises 3.46 million sentences and is approximately 1,000 times larger than those used in prior work.
- We introduce a selective unlearning framework, namely SUV, that precisely targets problematic segments responsible for verbatim memorization, making it possible for mitigating copyright risks over large-scale corpus while preserving model utility.
- Extensive experiments on both our large-scale dataset and public benchmarks (Wei et al., 2024; Dou et al., 2025) demonstrate that the proposed approach achieves a superior balance between effective unlearning and the maintenance of overall model performance.

## 2 Related Work

**Copyright Issues in LLMs.** Prior work has focused on examining the extent to which language models reproduce copyrighted text and how this behavior relates to the broader phenomenon of memorization. Several studies have employed various probing techniques, such as similarity measurement and masked token prediction, to evaluate verbatim reproduction and data extraction risks (Chang et al., 2023; Karamolegkou et al., 2023; D'Souza & Mimno, 2023; Hacohen et al., 2024; Nasr et al., 2023; Schwarzschild et al., 2024; Li et al., 2024; Liu et al., 2024c; Chen et al., 2024). Investigations have revealed that model capacity and training procedures play a significant role in memorization, with even state-of-the-art models outputting verbatim excerpts from their training data (Carlini et al., 2021; 2023; Biderman et al., 2023; Wei et al., 2024; Liu et al., 2024c). Concurrently, other works have introduced new evaluation metrics and defense assessments to better understand these copyright implications (Wei et al., 2024; Mueller et al., 2024; Chen et al., 2024). Further studies have explored detection methods and data curation techniques to mitigate memorization risks (Huang et al., 2024; Kim et al., 2023; Elangovan et al., 2021; Lee et al., 2022). Efforts to mitigate these issues have been broadly categorized into methods that enable models to forget specific training data associated with copyrighted content, i.e. unlearning (Liu et al., 2024b;d; Yao et al., 2023; Hans et al., 2024; Chen & Yang, 2023; Hans et al., 2024; Maini et al., 2024; Dou et al., 2025; Zhang et al., 2024), modifications at the decoding stage to steer generation away from potentially infringing text (Ippolito et al., 2023a; Xu et al., 2024b; Abad et al., 2024a), with additional enhancements explored via agents (Liu et al., 2024c) or RAG (Golatkar et al., 2024). Our method falls into the unlearning category, enabling copyright compliance for open-weight LLMs. Current unlearning methods fail to handle large-scale corpus while retaining the model performance. In contrast, SUV employs targeted elimination to unlearn specific content while preserving the model's overall knowledge and utility.

**Machine Unlearning in LLMs.** Machine unlearning seeks to selectively remove undesired knowledge from a pretrained model while preserving its overall capabilities (Cao & Yang, 2015; Hoofnagle et al., 2019; Bourtoule et al., 2021; Nguyen et al., 2022; Liu et al., 2024d;e; Golatkar et al., 2020a;b; 2021; Dukler et al., 2023). A central challenge is catastrophic forgetting, whereby unlearning targeted information inadvertently degrades untargeted knowledge and impairs model performance (Tarun et al., 2023; Xu et al., 2024a; Fan et al., 2024; Zhang et al., 2024). In the context of LLMs, existing mitigation strategies fall into

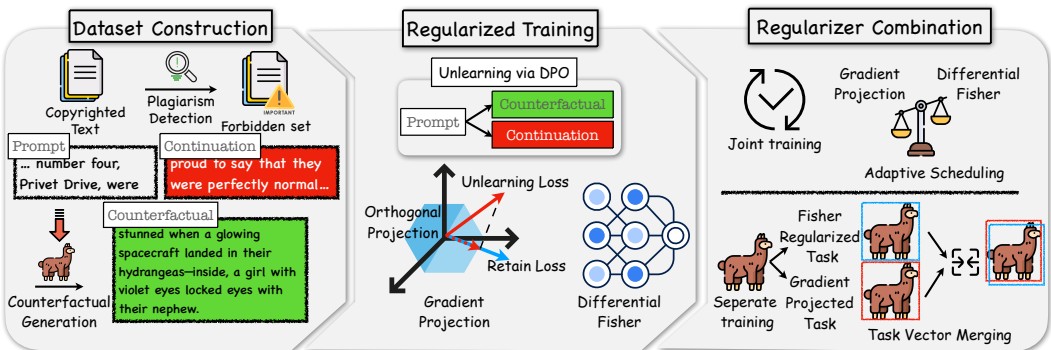

Figure 2: An overview of the SUV Framework

two broad categories: parameter-based unlearning (Jang et al., 2022; Yu et al., 2023; Zhang et al., 2023; Wu et al., 2023; Hase et al., 2023) and prompt-based unlearning (Madaan et al., 2022; Pawelczyk et al., 2023; Thaker et al., 2024; Muresanu et al., 2024; Liu et al., 2024a). Prompt-based methods avoid the computational expenses of retraining by steering model outputs at inference time, but they neither yield a permanently modified model nor fully guard against adversarial "jailbreak" attacks (Liu et al., 2025). In contrast, parameter-based approaches (Zhang et al., 2024; Jia et al., 2024) produce a persistent and robust unlearned model, with the added benefit of preserving other learned knowledge. Despite these advances, it remains challenging to remove verbatim memorization of copyrighted materials in large-scale datasets without affecting performance. SUV employs a selective approach that precisely targets unwanted content while preserving learned representations, making it suitable for large-scale unlearning scenarios.

## 3 Methodology

We introduce SUV, a novel training framework to enable large language models (LLMs) to selectively unlearn extensive portions of book data while preserving performance on unrelated tasks (see Figure 2). SUV comprises three key stages: (1) **Dataset Construction** (Section 3.2): We utilize an efficient sliding-window mechanism to identify verbatim excerpts from the specified texts, with each book processed in just a few minutes. Then, based on the verbatim excerpts, we prepare a **counterfactual** dataset for DPO that helps reduce influence on unrelated tasks while unlearning the texts. (2) **Regularized Training** (Section 3.3): During the DPO training, to minimize the adverse impact of unlearning on other tasks, we regulate the direction of parameter updates using **gradient projection** and select the most relevant parameters based on **Fisher information**. (3) **Regularization Combination** (Section 3.4): Since there might be interference between the two regularization methods, we propose two variants on combining them to better balance both unlearning performance and utility.

### 3.1 Problem Definition

Let $f_\theta$ denote a language model with parameters $\theta$, where $f_\theta(x)$ denotes the probability of generating the sequence $x$ under the model. We collect a set of books to be unlearned, which we denote as $\mathcal{D}_B$, named as the *book dataset*. We denote by $\mathcal{D}_U$ the unrelated data, and our training data consists of $\mathcal{D} = \mathcal{D}_B \cup \mathcal{D}_U$. The goal of unlearning is to update the model's parameters from $\theta$ to $\theta'$ by solving the following optimization problem:

$$\min_{\theta'} \ \mathbb{E}_{x \in \mathcal{D}_B}\big[f_{\theta'}(x)\big] \quad \text{s.t.} \quad \big|\mathcal{L}_U(\theta) - \mathcal{L}_U(\theta')\big| \leq \epsilon, \tag{1}$$

where the loss on $\mathcal{D}_U$ is defined as $\mathcal{L}_U(\theta) = \mathbb{E}_{x \in \mathcal{D}_U}\big[\mathcal{L}(x;\theta)\big]$, where $\mathcal{L}(x;\theta)$ denotes the loss function (negative log-likelihood) on input $x$ for model parameter $\theta$. Here, $\epsilon > 0$ is a pre-specified tolerance parameter that controls the acceptable degradation on the performance over $\mathcal{D}_U$. This formulation minimizes the generation probability on the forbidden dataset while ensuring that the performance on unrelated tasks does not degrade significantly.

## 3.2 Dataset Construction

We construct a DPO dataset that meets the requirement of the above optimization problem (cf. Equation 1) . A DPO dataset consists of preference triples $(x^{(p)}, y^{(w)}, y^{(l)})$, where $x^{(p)}$ is the prompt, $y^{(w)}$ is the preferred continuation, and $y^{(l)}$ is the less-preferred continuation. We divide the dataset construction into two steps: (1) **Plagiarism Detection**, which verifies the subsequences in $\mathcal{D}_B$ against the model's memory, ensures $x^{(p)}$ is indeed part of $f_\theta$'s training set, then constructs $x^{(p)}$ and $y^{(l)}$ pairs, and (2) **Counterfactual Generation**, which generates $y^{(w)}$ as a counterfactual continuation to $x^{(p)}$.

**Plagiarism Detection.** Each book $x \in \mathcal{D}_B$ is tokenized into a sequence $x = (x_1, x_2, \ldots, x_T)$, $x_t \in \mathcal{V}$, $t \in \{1, \ldots, T\}$, where $\mathcal{V}$ is the vocabulary. We partition the sequence into $K$ subsequences using segmentation indices $1 = k_0 < k_1 < \cdots < k_K = T$; the $i$-th segment is $(x_{k_i}, x_{k_i+1}, \ldots, x_{k_{i+1}})$, where $i \in \{0, \ldots, K-1\}$. In our experiments, for convenience, we employ fixed-length segmentation, such that for all $0 \le i < K$, $k_{i+1} - k_i$ is fixed. With sufficiently long segments, we approximate

$$f_{\theta'}(x) = \prod_{i=0}^{K-1} f_{\theta'}(x_{k_i}, \ldots, x_{k_{i+1}} \mid x_1, \ldots, x_{k_i-1}) \approx \prod_{i=0}^{K-1} f_{\theta'}(x_{k_i}, \ldots, x_{k_{i+1}}). \tag{2}$$

We define the *forbidden set* $\mathcal{D}_F$ as the set of segments from $\mathcal{D}_B$ with notably high probability:

$$\mathcal{D}_F = \left\{ (x_{k^*}, \ldots, x_{k^*+1}) \,\middle|\, 0 \le k^* \le K-1, \, f_\theta(x_{k^*}, \ldots, x_{k^*+1}) \ge \epsilon \right\}, \tag{3}$$

with a threshold $\epsilon \ge 0$. In practice, only a few segments per book are memorized, meaning that instead of training on the large $\mathcal{D}_B$, we can train on the considerably smaller $\mathcal{D}_F$ to accelerate the training process. To detect these memorized segments, we apply a sliding window over $x$. For a window starting at token $x_p$, we define the *prompt* as $x^{(p)} = (x_p, x_{p+1}, \ldots, x_{p+L_p-1})$, and the corresponding *original continuation* as $y^{(l)} = (x_{p+L_p}, x_{p+L_p+1}, \ldots, x_{p+L_p+L_c-1})$, where $L_p$ and $L_c$ denote the lengths of the prompt and continuation, respectively. A candidate continuation is generated by randomized sampling: $y^{(g)} \sim P_\theta(y \mid x^{(p)})$. If the similarity (e.g., measured by LCS or ROUGE-L) between $y^{(g)}$ and $y^{(l)}$ exceeds a threshold $\tau$, the segment is classified as belonging to $\mathcal{D}_F$.

**Counterfactual Generation.** For each prompt $x^{(p)}$ with its rejected continuation $y^{(l)}$, we generate a counterfactual (preferred) continuation $y^{(w)}$ using a counterfactual generation process. Specifically, we prompt the model $f_\theta$ with $x^{(p)}$ and instruct it to produce an alternative continuation in a different genre from $y^{(l)}$. This counterfactual continuation has both a low perplexity and a significant divergence from $y^{(l)}$. We detail the generation process in Appendix F.

### 3.3 Regularized Training

With the data constructed in Section 3.2, we can easily fine-tune the target language model with DPO. However, using the DPO dataset alone can reduce the model's likelihood of generating memorized content from $\mathcal{D}_B$, straightforward DPO training may compromise performance on the unrelated data $\mathcal{D}_U$. The core issue is that the gradient updates for unlearning can conflict with knowledge crucial for maintaining performance on $\mathcal{D}_U$. To address this, we introduce two key techniques: (1) **Gradient Projection**, which ensures the unlearning process does not degrade the model's existing capabilities, and (2) **Fisher-based Regularization**, which protects parameters highly sensitive for other tasks. We provide a detailed description of the techniques we used in Appendix A.

**Gradient Projection.** To ensure that unlearning updates do not conflict with preserving performance on $\mathcal{D}_U$, we employ a *gradient projection* strategy. Let $\mathcal{L}_{DPO}$ be the DPO loss with gradient $g_p = \nabla \mathcal{L}_{DPO}$. We also define a *preservation gradient* $g_u$, for instance, by maintaining a sliding exponential moving average of gradients computed on $\mathcal{D}_U$.

When $\langle g_u, g_p \rangle < 0$, the unlearning gradient $g_p$ is in direct opposition to $g_u$, which would risk degrading performance on $\mathcal{D}_U$. To avoid this, we remove from $g_p$ the component that negatively aligns with $g_u$. Formally, we define the *projected unlearning gradient*:

$$
\tilde{g}_p = \begin{cases} g_p - \dfrac{\langle g_u, g_p \rangle}{\|g_u\|^2}\, g_u, & \text{if } \langle g_u, g_p \rangle < 0, \\ g_p, & \text{otherwise.} \end{cases} \tag{4}
$$

The final combined gradient for the update is then $g_{\text{proj}} = g_u + \tilde{g}_p$, which guarantees that the unlearning updates do not counteract the preservation gradient, thereby ensuring that the model's performance on $\mathcal{D}_U$ is preserved.

**Fisher Regularization.** While the safe update criterion defines which updates preserve performance on $\mathcal{D}_U$, we also need a practical mechanism to steer the optimizer away from harmful parameter changes. To this end, we introduce a *Fisher-based regularization* term, namely *Differential Fisher Importance*, formally

$$
\Delta F_i = \max\Big( F_i^{(\text{forb})} - F_i^{(\text{retain})}, \varepsilon \Big), \tag{5}
$$

where $\varepsilon > 0$ ensuring a lower bound, $F_i^{(\text{forb})}$ and $F_i^{(\text{retain})}$ be the Fisher information for parameter $i$ on $\mathcal{D}_F$ and $\mathcal{D}_U$, respectively. It penalizes large changes to parameters crucial for preserving knowledge on $\mathcal{D}_U$. A large $\Delta F_i$ indicates that parameter $i$ is more important for memorized content than for $\mathcal{D}_U$.

Let $\Delta \theta = \theta' - \theta$ be the parameter update. We then define a coordinate-wise Fisher penalty:

$$
\ell_i^{(\text{Fisher})} = \log\Big( \frac{(\Delta \theta_i)^2}{(\Delta F_i)^2} + 1 \Big),
$$

and aggregate them for all the updated parameter with hyperparameter $\lambda_{\text{fisher}}$ as the *Fisher regularization loss*: $L_{\text{freg}} = \lambda_{\text{fisher}} \sum_i \ell_i^{(\text{Fisher})}$. Here, a log function is incorporated to mitigate the risk of experiencing overflow errors or excessively small gradients, which could hinder the optimization process. Furthermore, by penalizing large updates to parameters critical for $\mathcal{D}_U$, this regularization nudges the optimizer toward updates preserving performance on unrelated tasks while reducing memorized content from $\mathcal{D}_B$.

### 3.4 Regularization Combination

Recall that we have two regularization methods. We now detail how to fuse them to create a unified framework that leverages the unique advantages of each approach. Here, we provide two variants for SUV: (1) SUV-AS, which is designed to preserve utility by jointly optimizing both regularization losses; and (2) SUV-TV, which maximizes copyright protection by isolating the effects of each regularization.

**Utility Preserving Training (SUV-AS).** In this variant, our goal is to prevent utility drop by leveraging both Fisher-based and gradient projection losses, as each plays a complementary role in protecting the parameters import to $\mathcal{D}_U$. Directly adding these losses is an intuitive way to maximize protection, with the gradient projection helping maintain stable update directions and the Fisher-based regularization constraining update magnitudes. However, the Fisher loss can dominate when other parts of the model are actively updated, making the optimization challenging. To overcome this, we introduce an adaptive scheduler for the Fisher regularization weight $\lambda_{\text{fisher}}$. Starting from a fixed initial value, $\lambda_{\text{fisher}}$ decays dynamically based on the training progress: a mild decay ($\rho_{\text{mild}} < 1$) is applied if the DPO loss decreases, while a severe decay ($\rho_{\text{severe}} \ll \rho_{\text{mild}}$) is used when no improvement is observed over $R$ consecutive steps. This dynamic balancing—termed SUV-AS —ensures that both losses contribute effectively to preserving the overall utility of the model throughout training. A detailed description of the scheduler is provided in Appendix G.

**Maximized Copyright Protection via Task Vector Merging (SUV-TV).** Alternatively, to achieve stronger copyright protection, we isolate the Fisher-based and projection-based

regularizations by training them separately. This separation is crucial because the Fisher-based task, which limits update magnitudes using differential Fisher information, and the projection-based task, which refines the update direction by projecting the unlearning gradient onto a stable preservation gradient, can potentially conflict when trained jointly. By handling each task independently, we ensure that the full strength of each regularization is preserved. After obtaining the two task-specific updates, $\Delta W_1 = W_1 - W_{\text{base}}$ *and* $\Delta W_2 = W_2 - W_{\text{base}}$ we merge them using a task vector (Ilharco et al., 2022): $W_{\text{merged}} = W_{\text{base}} + \Delta W_1 + \Delta W_2$. This method, referred to as SUV-TV, isolates the effects of each regularization so that the robust protection against copyright infringement is maximized without compromising the model's core capabilities.

**Discussion.** A theoretical trade-off is discussed in Appendix B regarding the unlearning effect and LLM utility. Our analysis reveals that SUV-AS achieves superior performance in LLM utility, while SUV-TV excels in unlearning. Empirical studies (see Section 4) further substantiate these findings. This outcome aligns with our expectations: in SUV-AS, the constraints imposed by gradient projection and Fisher regularization jointly influence the model, thereby amplifying the effect of unrelated data, whereas in SUV-TV, these constraints are applied independently across two models, which diminishes the impact of such data.

## 4 Experiments

### 4.1 Experiment settings

**Evaluation Datasets.** We evaluate SUV using two groups of benchmarks: **unlearning** and **utility** benchmarks. For assessing unlearning performance, we employ several datasets: (1) **CBP-500 (Copyrighted Books (Predominantly)-500)**, our dataset that is comprised of texts from 500 deduplicated and random famous books (predominantly copyrighted); (2) **CBP-50**, a randomly chosen subset of the CBP-500 dataset is used to ensure a fair comparison between SUV and baselines that cannot scale to 500 books; (3) **CotaEval NewsQA** (Wei et al., 2024), a publicly available benchmark that is comprised with texts from news articles; and (4) **SSU Books** (Dou et al., 2025), the 10 books used for training SSU, a baseline approach discussed below. We refer to Appendix E.1 for details on the datasets. For retaining the utility of language models, the **Alpaca** (Taori et al., 2023) dataset is used as the retain set of SUV, some baselines and ablation methods. For efficiency, we randomly sample 2,048 examples from **Alpaca** during training. To assess performance on unrelated tasks, we utilize **MMLU** (Hendrycks et al., 2021), a benchmark evaluating model knowledge across diverse subjects, and **CSQA** (Talmor et al., 2018), a benchmark with challenging multi-hop multiple-choice questions. Details can be found in Appendix E.3.

**Evaluation Metrics.** For unlearning benchmarks, we use ROUGE-L and LCS to measure text similarity and report the number of sentences with different ROUGE-L scores. For utility benchmarks, we follow the original dataset's accuracy criteria (Lin, 2004).

**Baselines.** We compare with: (1) the language model without unlearning (**Vanilla**); (2) Stable Sequential Unlearning (**SSU**) that iteratively removes restricted data effects (Dou et al., 2025); (3) Gradient Ascent (**GA**) which applies reverse optimization on undesirable content; and (4) Negative Preference Optimization (**NPO**) that minimizes a preference function for unwanted data (Zhang et al., 2024).

**Implementation Details.** SUV, baselines, and ablations are implemented with the Llama 3.1-8B model for fair comparison. More details can be found in Appendix E.4. Additional experiments on Qwen 3-8B and Qwen 3-14B models are also conducted to verify the effectiveness of SUV over different model architectures and sizes, as detailed in Appendix D.

### 4.2 Experimental Results

**Unlearning Performance.** Figure 3(a)–(c) presents the unlearning performance of the proposed methods along with various baselines, where lower ROUGE-L scores indicate better removal of verbatim copyrighted material. We evaluate all the methods on fixed-size chunks divided from the original books (for simplicity, let's just call them sentences) and report the number of sentences that exceed a certain ROUGE-L threshold. We analyze the unlearning

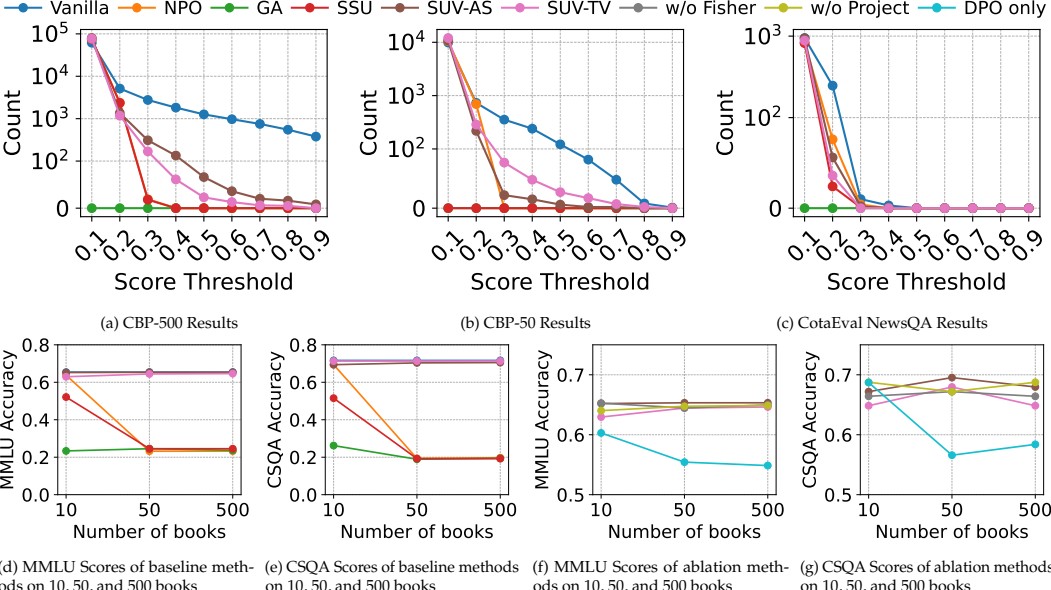

(a) CBP-500 Results  (b) CBP-50 Results  (c) CotaEval NewsQA Results

(d) MMLU Scores of baseline methods on 10, 50, and 500 books

(e) CSQA Scores of baseline methods on 10, 50, and 500 books

(f) MMLU Scores of ablation methods on 10, 50, and 500 books

(g) CSQA Scores of ablation methods on 10, 50, and 500 books

Figure 3: (a)–(c): The unlearning experiment results of baseline and SUV methods on the CBP-500, CBP-50, and CotaEval NewsQA dataset. (d)–(g): The utility experiment results of baseline, SUV, and ablation models trained on different datasets. Here, the x-axis is the number of books, where 10 is for SSU Books, 50 for CBP-50, and 500 for CBP-500.

performance by calculating the ratio of the sentences with ROUGE-L scores exceeding specific thresholds (the lower the better unlearning performance). The ratios obtained by the proposed method and the Vanilla model are compared. Specifically, on the CBP-500 dataset, at the 0.2 threshold, the Vanilla model registers 5,133 sentences while the proposed method (using SUV-TV) retains only 1,176 sentences, corresponding to approximately 23% of the Vanilla count. For the 0.3 threshold, the proposed method's count is 169 (versus 2,762 for the Vanilla model), which is only about 6% of the original. This gap further widens at the 0.5 threshold, where the proposed method retains roughly 1.8% (23 versus 1,264 sentences) of the Vanilla count, and at the 0.7 threshold, only about 0.8% (6 versus 751 sentences) remains. These results demonstrate that the proposed approach can selectively remove copyrighted content down to nearly 1/100 or less of the original amount when more stringent thresholds are applied. We also notice that the baselines, such as SSU and GA, tend to aggressively reduce the number of sentences above the given thresholds; however, this reduction is achieved at the expense of utility (detailed below).

**Utility Performance.** While some baselines may achieve low ROUGE-L counts at higher thresholds, they often do so at the cost of downstream task performance degradation. For example, methods like SSU and GA, despite having slightly lower unlearning metrics, experience significant degradation in utility metrics. For example, their CSQA scores drop notably compared to the Vanilla model's 0.7166. We further show that they have made the model unusable with a case study detailed in Section 4.2. In contrast, our SUV-AS approach consistently maintains utility scores close to the vanilla model (0.7093 for CSQA and 0.6622 for MMLU on CotaEval NewsQA), ensuring that the aggressive unlearning does not compromise overall performance. This dual-method framework—where SUV-TV excels at targeted content removal and SUV-AS preserves utility—underscores the practical advantage of the proposed approach.

**Ablation Study.** We conduct an ablation to verify several design choices in SUV: (1) **SUV-TV** vs. **SUV-AS**: We compare two regularization merging techniques. SUV-AS uses an adaptive scheduler to merge regularization signals, while SUV-TV relies on a task vector approach. (2) **w/o Fisher**: This variant evaluates SUV using only the gradient projection regularization, omitting the Fisher regularization. (3) **w/o Project**: Here, we assess the performance when only Fisher regularization is applied, without the gradient projection component. (4) **DPO only**: In this setup, SUV is trained solely with the DPO objective, without any form of regularization. We compare several ablated variants of our method in Table 1. As

| Method / Dataset | CotaEval | | | | | | CBP-500 | | | | | |
|---|---|---|---|---|---|---|---|---|---|---|---|---|
| **Metrics** | ROUGE-L (↓) | | LCS (↓) | | Utility (↑) | | ROUGE-L (↓) | | LCS (↓) | | Utility (↑) | |
| | Avg | Max | Avg | Max | MMLU | CSQA | Avg | Max | Avg | Max | MMLU | CSQA |
| Vanilla model | 17.29 | 60.24 | 334.94 | 635 | .6542 | .7166 | 11.54 | 100.00 | 19.60 | 99 | **.6542** | .7166 |
| SUV-TV | **13.58** | **26.58** | **306.01** | **438** | .6555 | **.7199** | 11.64 | **86.00** | 14.40 | **85** | .6463 | .7133 |
| SUV-AS | 15.09 | 31.80 | 334.62 | 450 | **.6622** | .7093 | 11.90 | 98.04 | 13.31 | 95 | .6533 | .7052 |
| w/o Fisher | 13.67 | 26.86 | 306.09 | 439 | .6533 | .7166 | **9.02** | 98.92 | 14.05 | 99 | .6497 | **.7199** |
| w/o Project | 13.87 | 26.98 | 314.48 | 459 | .6504 | .7052 | 9.94 | 100.00 | 15.44 | 99 | .6488 | .7052 |
| DPO only | 15.74 | 31.42 | 343.17 | 472 | .6456 | .6994 | 12.02 | 98.08 | **12.96** | 99 | .5486 | .5839 |

Table 1: The unlearning and utility experiment results of ablations. ↓ indicates lower is better, and ↑ indicates higher is better. The best results are marked in **bold**.

| | |
|---|---|
| **Prompt** | Mr. and Mrs. Dursley, of number four, Privet Drive, were proud to say that they were |
| **Original** | perfectly normal, thank you very much. They were the last people you'd expect to be involved in anything strange or mysterious, because they just didn't hold with such nonsense. |
| **Vanilla** | perfectly normal, thank you very much. They were the last people you'd expect to be involved in anything strange or mysterious, because they just didn't hold with such nonsense. |
| **SUV** | perfectly normal, boring people, if you don't count all the money they made from the wizarding world. But little did they know, their life was about to change forever. |
| **GA** | ,,,,,,,,,,,,,,,,,,,,,,,,,,,,, ,, (,,,,,,,,,,,,,,, ,, (,,, (gibberish) |
| **SSU** | """""""""""""""""""""""""""""""""""""""""""""""""" (gibberish) |
| **NPO** | nYou. I http://www.thecontaminated and but if a person http Achilles tendinitis a I and. He the buy cialis on The Pneumococcal of. " This This |

Table 2: Case study of SUV to generate non-copyright-infringing continuations for a piece of copyrighted text from Harry Potter. Red indicates overlaps with the original continuation.

expected, while SUV-AS shows a modest drop in unlearning performance, it achieves the best utility performance on MMLU (e.g., 0.663 vs. 0.655) while being on par with SUV-TV on CommonsenseQA. In contrast, SUV-TV remains the optimal choice when prioritizing unlearning effectiveness (for instance, achieving an LCS average of 15.38 on CotaEval). The performance drop observed when either regularization is removed (e.g., "w/o Fisher" leading to an LCS average above 39) highlights the importance of combining both techniques. Moreover, the "DPO only" variant demonstrates the poorest performance in both unlearning and utility, likely because the DPO objective deviates from the unlearning focus. Overall, these ablation results confirm the robustness and effectiveness of our design choices in SUV.

Considering both Figure 3 (d) and (f), or (e) and (g), we also observe that although our DPO training process without regularization retains the least utility among the ablation study, it still significantly outperforms the baselines. Specifically, after applying our DPO-only method on CBP-50, the model achieves 0.554 on MMLU, which is more than twice the number SSU achieves (0.245). Still, this result is achieved without using an additional retain set. This highlights the strong utility preservation enabled by unlearning through a diversified, model-generated counterfactual continuation, as well as the targeted plagiarism detection.

**Scalability Study** We further evaluate the scalability of SUV on four increasingly large datasets–NewsQA, SSU Books, CBP-50, and CBP-500–and present the results in Figure 3. Notably, on the smaller NewsQA set, SUV reduces the verbatim generation of forbidden text by over 30% while retaining a utility score (MMLU) within 2 points of the vanilla model. As we scale to the medium-sized CBP-50, this reduction surpasses 40%, with MMLU remaining above 0.65. Even on the largest CBP-500 dataset, SUV continues to effectively unlearn copyrighted content—cutting LCS (lower is better) by nearly half—while preserving utility scores (e.g., around 0.64 on MMLU). These improvements across progressively larger datasets confirm that SUV scales effectively, maintaining robust unlearning capabilities without compromising downstream performance.

**Case Study.** Table 2 presents a case study from a copyrighted narrative, demonstrating the effectiveness of SUV in generating non-infringing continuations. The original continuation provides a well-known description of the Dursleys, which the Vanilla model replicates verbatim. In contrast, SUV produces a transformed continuation that maintains narrative coherence—describing the Dursleys as "perfectly normal, boring people" with a subtle twist hinting at hidden fortunes—without replicating the copyrighted text. Meanwhile, both GA and SSU yield outputs that degrade into gibberish, and NPO produces an incoherent sequence of words. This case study clearly illustrates that SUV successfully removes copyrighted content while preserving the story's logical progression and utility.

## 5 Conclusion

In this work, we introduced SUV, a selective unlearning framework that efficiently mitigates the verbatim memorization of copyrighted content in LLMs. By leveraging DPO together with projected gradient techniques and Fisher information regularization, SUV precisely targets and removes unwanted content while preserving the integrity of the model's learned knowledge. Our extensive experiments on a large-scale dataset of 500 books, as well as on public benchmarks, demonstrate that SUV achieves a superior balance between effective unlearning and maintaining overall model utility. This scalability and precision make the proposed approach a promising solution for addressing copyright risks in real-world LLM applications.

## Ethics Statement

In this work, our primary goal is to safeguard the intellectual property of authors and publishers against AI-driven copyright infringement. In an era of widespread information access, protecting copyrighted materials has become increasingly challenging. The proposed approach leverages cutting-edge technologies to identify and eliminate unauthorized reproductions of copyrighted text from generative models, while rigorously upholding the rights of content creators. To this end, we have implemented measures that ensure the proposed method remains both respectful of intellectual property rights and anchored in principles of fairness and responsibility.

A critical aspect of our research is the evaluation of copyright infringement, which necessitates training our model on the full versions of books. This comprehensive strategy is essential to capture the nuances of memorization and to develop effective unlearning techniques. Due to legal and ethical considerations, we do not plan to publicly release our complete dataset. Instead, we will provide a detailed list of the book titles used in our study upon request, allowing interested parties to obtain these works through academic or public libraries.

Our use of copyrighted materials is strictly non-commercial and transformative. In repurposing these works to derive new insights into copyright protection, our research contributes to academic scholarship without diminishing the market value of the original texts. In doing so, we adhere to U.S. copyright law (17 U.S. Code § 107 - Limitations on exclusive rights: Fair use), and our approach qualifies as fair use. Specifically, despite the provisions of sections 106 and 106A, fair use permits the reproduction of copyrighted works for purposes such as criticism, commentary, news reporting, teaching, scholarship, or research.

Limited excerpts of copyrighted text may appear in figures, tables, or examples solely for research and analysis, with proper attribution given to the original authors and publishers. These measures, along with our steadfast commitment to ethical practices and non-commercial use, underscore our dedication to protecting intellectual property rights in the digital age.

## Acknowledgements

This work was funded in part by the National Science Foundation under Grant No. IIS-2141037.

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

## A  Preliminary

### A.1  Causal Language Model

In a **causal language model**, we predict each token based on its preceding tokens, thereby maximizing the sequence likelihood. Equivalently, we minimize the negative log-likelihood loss. Formally, given a sequence $x = (x_1, x_2, \ldots, x_T)$, the loss function is defined as:

$$\mathcal{L}(x; \theta) = -\frac{1}{T} \sum_{t=\tau}^{T} \log P(x_t \mid x_{<\tau}, x_\tau, \ldots, x_{t-1}; \theta)$$

, where $x_{<\tau}$ is the prefix. Here, $P(x_t \mid x_{<\tau}, x_\tau, \ldots, x_{t-1}; \theta)$ is the probability of token $x_t$ given the preceding context, as predicted by the model $f_\theta$ after applying the Softmax function over the output logits.

### A.2  Fisher Information.

For a causal language model $f_\theta$ generating a sequence $x$, the Hessian of the negative log-likelihood,

$$\mathbf{H}_\theta(x) = -\nabla_\theta^2 \log P(x; \theta),$$

characterizes the local curvature of the loss landscape, thereby quantifying parameter sensitivity. Under standard regularity conditions (ensuring smoothness and the interchangeability of differentiation and expectation), the expected Hessian is equivalent to the **Fisher information matrix** (Fisher, 1922):

$$F_\theta = \mathbb{E}_{x \sim f_\theta} \left[ \nabla_\theta \log P_\theta(x)^\top \nabla_\theta \log P_\theta(x) \right] = -\mathbb{E}_{x \sim f_\theta} \left[ \nabla_\theta^2 \log P_\theta(x) \right]. \tag{6}$$

Since storing $F_\theta$ directly requires $O(|\theta|^2)$ memory, a common remedy (Kirkpatrick et al., 2017; Matena & Raffel, 2022) is to use the empirical Fisher approximation,

$$\hat{F}_\theta = \frac{1}{N} \sum_{i=1}^{N} \left( \nabla_\theta \log P(x^{(i)}; \theta) \right)^2,$$

where $x^{(1)}, \ldots, x^{(N)}$ are $N$ samples drawn i.i.d. from the training dataset, and the per-sample gradients are efficiently computed via backpropagation.

### A.3  Gradient Projection Method

Gradient Projection Method (Farajtabar et al., 2019; Saha et al., 2021) is an optimization technique for solving constrained problems by ensuring that the update direction remains within a feasible set $\mathcal{C}$. Instead of the standard update rule, one first computes an unconstrained step and then projects it onto a feasible set $\mathcal{C}$:

$$\theta^{(t+1)} = \Pi_\mathcal{C} \left( \theta^{(t)} - \eta \nabla \mathcal{L}(\theta^{(t)}) \right), \quad \text{with} \quad \Pi_\mathcal{C}(x) = \arg \min_{y \in \mathcal{C}} \|x - y\|^2. \tag{7}$$

In multi-task learning, given task gradients $g_i$, one may choose

$$\mathcal{C} = \bigcap_i \{ d \in \mathbb{R}^{|\theta|} \mid \langle d, g_i \rangle \geq 0 \},$$

a strategy that similarly applies to unlearning tasks by balancing removal of undesired information with preserving existing knowledge.

### A.4  Direct Preference Optimization (DPO)

DPO (Rafailov et al., 2024) fine-tunes language models to align with human preferences. Consider a preference triple $(x_p, y_w, y_l)$, where $x_p$ is an input (or context), $y_w$ is the preferred response, and $y_l$ is the less-preferred response. Let $P_\theta(y \mid x_p)$ denote the probability of

generating response $y$ under the fine-tuned model, and $P_{\text{ref}}(y \mid x_p)$ the corresponding probability under a reference model. The DPO loss is defined as

$$\mathcal{L}_{\text{DPO}}(P_\theta; P_{\text{ref}}) = -\mathbb{E}_{(x_p, y_w, y_l) \sim \mathcal{D}} \left[ \log \sigma \left( \beta \, \Delta \log P_\theta(x_p) \right) \right], \tag{8}$$

where

$$\Delta \log P_\theta(x_p) = \log \frac{P_\theta(y_w \mid x_p)}{P_{\text{ref}}(y_w \mid x_p)} - \log \frac{P_\theta(y_l \mid x_p)}{P_{\text{ref}}(y_l \mid x_p)},$$

$\beta > 0$ is a hyperparameter, and $\sigma(\cdot)$ is the sigmoid function.

## B SUV Analysis

We provide the following analysis of SUV and its variants on their ability to unlearn as well as preserve utility.

**Analysis 1 (Utility is preserved within SUV).** To ensure that utility is preserved, we prove that the overall parameter update is bounded. A second-order Taylor expansion of the retention loss $\mathcal{L}_U$ at $W_{\text{base}}$ yields

$$\Delta \mathcal{L}_U \approx \nabla \mathcal{L}_U^\top \Delta W + \frac{1}{2} \Delta W^\top H_U \Delta W \leq \frac{\gamma}{2} \|\Delta W\|^2, \tag{9}$$

where $H_U$ is the Hessian of $\mathcal{L}_U$ with $\|H_U\| \leq \gamma$ and $\Delta W = \Delta W_1 + \Delta W_2$ is the overall update. This bound guarantees that the degradation in performance is tightly controlled.

**Analysis 2 (SUV-AS preserves utility better.)** Assume that both training strategies constrain the overall update $\Delta W$ so that the increase in the utility loss satisfies $\Delta L_U \approx \nabla L_U^\top \Delta W + \frac{1}{2} \Delta W^\top H_U \Delta W \leq \frac{\gamma}{2} \|\Delta W\|^2$. Then, the joint training approach in SUV-AS yields a strictly lower increase in $L_U$ than the decoupled training in SUV-TV. In particular, there exists a constant $\eta > 0$ such that $\Delta L_U^{\text{AS}} \leq \Delta L_U^{\text{TV}} - \eta$, where $\Delta L_U^{\text{AS}}$ and $\Delta L_U^{\text{TV}}$ denote the increments in $L_U$ for the joint and decoupled methods, respectively.

**Analysis 3 (SUV-TV yields better unlearning performance).** Under the bounded-update assumption for the retention loss, the decoupled training strategy in SUV-TV yields a more effective unlearning update than the joint-training strategy. In particular, there exists a constant $\xi > 0$ such that $\Delta P_B^{\text{TV}} \leq \Delta P_B^{\text{AS}} - \xi$, where $\Delta P_B^{\text{TV}}$ and $\Delta P_B^{\text{AS}}$ denote the decreases in the generation probability on the forbidden dataset $\mathcal{D}_B$ under the taskvector and joint-training approaches, respectively.

In what follows, we consider a base parameter vector $W_{\text{base}} \in \mathbb{R}^d$ and a small update $\Delta W \in \mathbb{R}^d$. Our analysis leverages a second-order Taylor expansion of the relevant loss functions around $W_{\text{base}}$.

### B.1 Notation and Preliminaries

**Loss Functions and Taylor Expansion.** For the *retention loss* on the unrelated data $\mathcal{D}_U$, denote the loss by $\mathcal{L}_U : \mathbb{R}^d \to \mathbb{R}$. A second-order Taylor expansion at $W_{\text{base}}$ yields

$$\Delta \mathcal{L}_U = \mathcal{L}_U(W_{\text{base}} + \Delta W) - \mathcal{L}_U(W_{\text{base}}) \approx \nabla \mathcal{L}_U^\top \Delta W + \frac{1}{2} \Delta W^\top H_U \Delta W,$$

where $H_U$ is the Hessian of $\mathcal{L}_U$ and it is assumed that $\|H_U\| \leq \gamma$ for some $\gamma > 0$. Similarly, for the *utility loss* (denoted $L_U$) defined on the relevant data, we have

$$\Delta L_U \approx \nabla L_U^\top \Delta W + \frac{1}{2} \Delta W^\top H_U \Delta W.$$

**Update Decomposition.** We decompose the overall update as

$$\Delta W = \Delta W_1 + \Delta W_2,$$

where:

- $\Delta W_1$ is the Fisher-based update.
- $\Delta W_2$ is the projection-based update, which is constructed to nearly cancel the linear term (i.e., $\nabla \mathcal{L}_U^\top \Delta W \approx 0$).

Under the assumption of near-orthogonality between $\Delta W_1$ and $\Delta W_2$, we have

$$\|\Delta W\|^2 \approx \|\Delta W_1\|^2 + \|\Delta W_2\|^2.$$

**Strategies.** In the decoupled strategy (referred to as SUV-TV), the overall update is given by

$$\Delta W^{\mathrm{TV}} = \Delta W_1 + \Delta W_2.$$

In the joint-training strategy (referred to as SUV-AS), the update is formed by simultaneously optimizing a Fisher-based regularization and a gradient projection. Denote by $g_u$ the preservation gradient (computed on $\mathcal{D}_U$) and by $g_p$ the unlearning gradient. The projection of $g_p$ is defined as

$$\tilde{g}_p = \begin{cases} g_p - \dfrac{\langle g_u, g_p \rangle}{\|g_u\|^2} \, g_u, & \text{if } \langle g_u, g_p \rangle < 0, \\ g_p, & \text{otherwise.} \end{cases}$$

Then, the joint-training update is

$$\Delta W^{\mathrm{AS}} = \alpha \, g_u + \tilde{g}_p,$$

with some scaling factor $\alpha > 0$.

### B.2 Analysis 1: Bounded Retention Loss

For both the SUV-TV and the SUV-AS methods, the projection-based update $\Delta W_2$ is designed to nearly cancel the linear term in the Taylor expansion, so that

$$\nabla \mathcal{L}_U^\top \Delta W \approx 0.$$

Thus, the change in retention loss is dominated by the quadratic term:

$$\Delta \mathcal{L}_U \approx \frac{1}{2} \Delta W^\top H_U \Delta W \leq \frac{\gamma}{2} \|\Delta W\|^2.$$

With the near-orthogonality assumption,

$$\|\Delta W\|^2 \approx \|\Delta W_1\|^2 + \|\Delta W_2\|^2.$$

In particular, in the SUV-AS method, if the Fisher-based update satisfies $\|\Delta W_1\| \leq B_F$ and the projection-based update satisfies $\|\Delta W_2\| \leq B_P$, then

$$\Delta \mathcal{L}_U \leq \frac{\gamma}{2} \left( B_F^2 + B_P^2 \right).$$

### B.3 Analysis 2: Utility Preservation

In the joint-training strategy (SUV-AS), the simultaneous optimization of Fisher-based regularization and gradient projection yields an update

$$\Delta W^{\mathrm{AS}} = \alpha \, g_u + \tilde{g}_p,$$

where the gradient projection ensures alignment with the preservation gradient $g_u$ so that the linear term $\nabla L_U^\top \Delta W^{\mathrm{AS}}$ is minimized (or non-positive). In contrast, the decoupled strategy (SUV-TV) computes

$$\Delta W^{\mathrm{TV}} = \Delta W_1 + \Delta W_2,$$

without joint alignment, leading to a potential misalignment such that

$$\nabla L_U^\top \Delta W^{\mathrm{TV}} \geq \nabla L_U^\top \Delta W^{\mathrm{AS}} + \eta_1,$$

for some $\eta_1 > 0$. Applying the Taylor bound for the utility loss:

$$\Delta L_U^{\mathrm{AS}} \le \nabla L_U^\top \Delta W^{\mathrm{AS}} + \frac{\gamma}{2}\|\Delta W^{\mathrm{AS}}\|^2,$$

and

$$\Delta L_U^{\mathrm{TV}} \le \nabla L_U^\top \Delta W^{\mathrm{TV}} + \frac{\gamma}{2}\|\Delta W^{\mathrm{TV}}\|^2,$$

and assuming similar overall update magnitudes, it follows that

$$\Delta L_U^{\mathrm{AS}} \le \Delta L_U^{\mathrm{TV}} - \eta,$$

for some constant $\eta > 0$. Hence, the joint-training strategy better preserves utility.

### B.4 Analysis 3: Stronger Unlearning Performance

For unlearning performance, we compare the decrease in the generation probability on the forbidden dataset $\mathcal{D}_B$. In the joint-training approach, interference between the two components may arise. Denote the interference term by $\eta_0 > 0$ so that the effective joint update is

$$\Delta W^{\mathrm{AS}} = \Delta W_1 + \Delta W_2 - \Delta I,$$

with $\|\Delta I\| \ge \eta_0$. In contrast, the decoupled strategy computes

$$\Delta W^{\mathrm{TV}} = \Delta W_1 + \Delta W_2.$$

Approximating the effect on the forbidden dataset via a first-order Taylor expansion,

$$\Delta P_B \approx \nabla P_B^\top \Delta W,$$

the interference in $\Delta W^{\mathrm{AS}}$ implies that there exists a constant $\xi > 0$ such that

$$\nabla P_B^\top \Delta W^{\mathrm{TV}} \le \nabla P_B^\top \Delta W^{\mathrm{AS}} - \xi.$$

Thus, the decoupled strategy achieves a strictly stronger reduction in the forbidden generation probability:

$$\Delta P_B^{\mathrm{TV}} \le \Delta P_B^{\mathrm{AS}} - \xi.$$

∎

## C Methodological Discussions

### C.1 Contribution

While regularised training techniques such as Fisher-information weighting and gradient projection are well known, recent additive approaches (Liu et al., 2024b) primarily sum separate "forget" and "retain" losses. By contrast, SUV integrates Direct Preference Optimisation (DPO) with two parameter-aware regularisers designed for copyright-sensitive, large-scale corpora:

- **Gradient-projection regularisation** removes the component of the unlearning gradient that would otherwise damage retained knowledge.
- **Differential-Fisher regularisation** constrains updates to parameters most critical for downstream utility.

This integrated design yields targeted forgetting and improved utility retention relative to additive baselines. Empirically, baselines such as GA and NPO are evaluated both for unlearning efficacy and for preservation of downstream ability, revealing that approaches which reduce similarity can still be impractical if they collapse utility; SUV maintains a favourable trade-off on both axes.

## C.2 Advantages over Prior Copyright-Protection Methods

**Theoretical Advantages.** SUV improves upon decoding-based and agent-based defences by embedding protection in the model rather than external control layers (which can be removed). It performs *selective* unlearning by pinpointing memorised snippets via sliding-window plagiarism detection and counterfactual generation, enhancing scalability and utility. Analysis bounds the net parameter update $\Delta W$, implying a small retention-loss increase on unrelated data. Further, joint optimisation in SUV-AS yields a strictly smaller increase in utility loss than decoupled training, while SUV-TV's separate task-vector merge achieves a strictly larger reduction in forbidden-data generation probability than joint training.

**Empirical Advantages.** On CBP-500 at ROUGE-L $\geq 0.5$, SUV-TV reduces high-similarity generations from 1 264 (vanilla) to 23 ($\approx 1.8\%$), whereas GA and SSU collapse utility. Despite aggressive removal, SUV-AS preserves downstream accuracy nearly intact (e.g., CSQA within 0.007 of vanilla, MMLU within 0.008), while GA/NPO suffer larger degradations. Case-study outputs demonstrate that SUV continuations remain fluent and coherent, in contrast to GA/SSU (gibberish) and NPO (incoherence).

## C.3 Variant Selection Guidance (SUV-AS vs. SUV-TV)

SUV-AS jointly applies forgetting and utility-preserving objectives at every update, effectively tethering the model and limiting drift. SUV-TV isolates the forgetting and utility updates and merges their task vectors; this maximises removal but leaves a slightly larger residual drift, explaining its stronger protection with a small additional utility cost. Thus, **SUV-AS** (joint optimisation) prioritises utility preservation and is suited for production scenarios requiring broad competence. **SUV-TV** (task-vector merging) maximises removal with a modest additional utility cost, preferred when compliance requires the strongest possible excision. Both variants maintain usable capability, unlike baselines that can collapse general utility.

## C.4 Operational Similarity Targets for Compliance

Legal frameworks provide no bright-line overlap percentage below which works are automatically non-infringing; courts assess whether any remaining fragment captures the "heart" of the original. Practically, one should reduce similarity as far as possible while maintaining acceptable utility. On CBP-500 (SUV-TV variant): at ROUGE-L $\geq 0.3$ the overlap falls to $\approx 6\%$ of vanilla; at $\geq 0.5$ to $\approx 1.8\%$; and at $\geq 0.7$ under 1%. A ROUGE-L operating range of 0.3–0.5 provides a prudent balance.

## C.5 Differential Fisher on LoRA Adapters

Differential Fisher importance is computed only on trainable LoRA adapters attached to attention projections. Specifically, we (i) compute Differential Fisher Importance ($\Delta F$) only for the parameters introduced by LoRA and (ii) leave the frozen backbone untouched. This is due to:

1. **Localising $\Delta F$ to the trainable sub-space.** In our unlearning protocol, the backbone remains frozen; only the LoRA matrices appended to q_proj, k_proj, v_proj, and o_proj are updated. Fisher information is the expectation of gradient outer products, and frozen weights have identically zero gradients. Including them would therefore leave the optimisation objective unchanged while inflating memory and wall-clock cost. By restricting $\Delta F$ to the low-rank LoRA factors, we keep the penalty perfectly aligned with the space that can actually move, enabling single-node training on the full 500-book corpus.

2. **Empirical sufficiency.** The ablation already reported in Table 1 ("w/o Fisher") shows that removing the adapter-level Fisher term markedly increases residual verbatim overlap and degrades downstream accuracy, demonstrating that the localised penalty is both necessary and sufficient. Extending $\Delta F$ to parameters

that never receive gradients would add compute without affecting either copyright removal or utility.

3. **Theoretical parsimony.** Fisher information is additive across independent parameter blocks. Because the backbone block is constant ($\Delta\theta = 0$), its contribution to the Fisher penalty is provably zero. Evaluating or storing it would therefore offer no optimisation benefit.

4. **Choice of LoRA target-modules.** We follow the prevailing "attention-only" convention and place adapters (and hence $\Delta F$) on the four self-attention projections. This mask is recommended in the Hugging Face PEFT documentation, appears in community examples such as a frequently cited Stack Overflow thread, and is treated as the baseline configuration in recent research (Hayou et al., 2024).

In summary, limiting $\Delta F$ to the LoRA adapters is a principled efficiency–effectiveness trade-off rather than an arbitrary restriction, and the chosen adapter mask is fully mainstream in current literature and practice.

### C.6  Role of DPO-Only and Importance of Regularisers

The DPO-only variant yields a non-trivial utility benefit due to selective unlearning and counterfactual data construction. However, the full SUV (DPO + regularisers) is necessary for scaling unlearning to medium and large settings while maintaining strong utility guarantees; both components are essential for the overall effect.

### C.7  Regularisation Effects Across Data Scales

Observed variability across datasets is explained by scale and heterogeneity. CotaEval is small and homogeneous with high baseline accuracy, leaving limited headroom; CBP-500 is larger and more diverse, exposing consistent gains for SUV over baselines. Interactions between Fisher and projection gradients are scale-dependent:

- *Very small datasets (e.g., CotaEval):* Fisher and projection gradients typically align, as both target a few strongly memorised patterns.
- *Very large datasets (e.g., CBP-500):* Averaging over many examples diffuses both gradients, which again align in practice.
- *Medium scale (CBP-50):* The Fisher gradient reflects general model behaviour, while the projection gradient targets specific memorised snippets; these become nearly orthogonal, and naive summation slightly dilutes each effect.

Overall, SUV exhibits clear, robust improvements across scales, with modest fluctuations at medium scale explained by gradient geometry.

## D  Additional Experiments

### D.1  Extended backbone evaluation: Qwen-3 8B & 14B

In this appendix we extend our study to the Qwen-3 family, showing that SUV works consistently across model sizes and architectures. We gauge *forgetting* on the **CBP-500** benchmark and *utility* on MMLU and CSQA; all other settings follow the main paper.

**Unlearning Performance.** Figure 4 (a)–(b) plots, for each ROUGE-L threshold, how many copyrighted sentences survive after unlearning. Vanilla Qwen-3 models retain far more text than either SUV variant—SUV-AS or the more aggressive SUV-TV. On Qwen-3 8B, a threshold of 0.2 leaves the vanilla model with 3 735 high-overlap sentences, while SUV-AS keeps 1 219 (32 %) and SUV-TV only 792 (21 %). Raising the threshold to 0.5 widens the gap: vanilla still memorises 764 sentences, whereas both SUV variants cut the number to under 45 ($\leq$ 6%). The 14B backbone shows the same pattern—SUV-TV drives the count below 10 % of vanilla for thresholds above 0.4, with SUV-AS close behind. Thus, SUV-TV delivers the strongest forgetting, and SUV-AS provides a slightly milder—but still substantial—alternative.

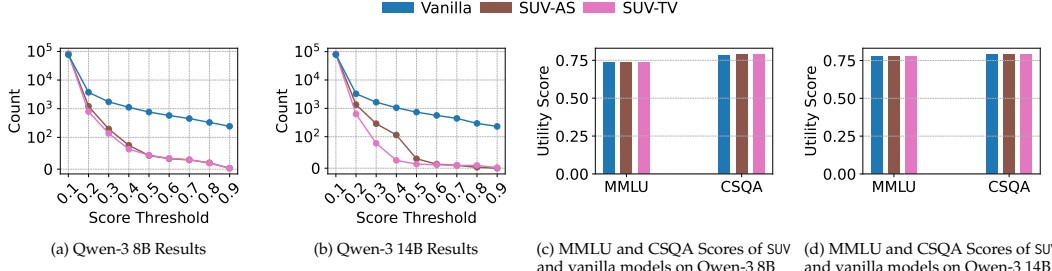

Figure 4: (a)–(b): The unlearning experiment results of vanilla and SUV models on the CBP-500 dataset conducted on Qwen-3 8B and 14B. (c)–(d): The utility experiment results of vanilla and SUV models on Qwen-3 8B and 14B.

**Utility Performance.** Figure 4 (c)–(d) reports accuracy on MMLU and CSQA. Despite the large reduction in memorised content, both SUV variants stay within one–two percentage points of the vanilla baseline. For Qwen-3 8B, vanilla scores 0.7366 / 0.7832 on MMLU / CSQA; SUV-AS yields 0.7369 / 0.7901 and SUV-TV 0.7365 / 0.7881. Results for the 14B model are nearly identical. In contrast, baselines such as GA or SSU typically lower ROUGE-L counts only by paying double-digit drops in accuracy. The two-stage design of SUV therefore achieves a desirable *forgetting-without-regretting* trade-off: SUV-TV maximises compliance by almost eliminating memorised passages, while SUV-AS offers a lighter-touch option that preserves virtually all task utility.

Because larger models tend to memorise more, the stability of SUV with increasing model size is critical. The extended Qwen results show that selective counterfactual generation plus regularisation remains effective as memorisation intensifies, delivering both strong removal and preserved utility.

### D.2 SSU Scalability and Comparison Feasibility

SSU (Dou et al., 2025) is fundamentally a sequential unlearning procedure, where each unlearning "step" builds upon the previous one. In our exact replication of Dou et al. (2025) setup, once SSU unlearns beyond a subset of CBP-50, the model's capacity as a language model collapses, so further training is neither informative nor useful. Table 3 reports the evaluation results at successive SSU checkpoints (each on CBP-50), following the original settings exactly. We observe that by Checkpoint 3 the model is already effectively non-functional as an LM, and by Checkpoints 4 and 5 all perplexities are infinite and accuracies have dropped to zero. Continuing to train on a larger subset (e.g., CBP-100 or CBP-500) would only exacerbate this collapse, at a very high computational cost, with no remaining language modeling ability to compare meaningfully against SUV.

Given this collapse, running SSU on larger subsets (e.g., CBP-500) yields no meaningful language-model capability for comparison. Therefore, we believe using CBP-50 for SSU is more than enough to provide a valid comparison with SUV on CBP-500.

### D.3 Nuanced Interpretation of Baseline Utility Degradation

In the main text we report the utility results of the baselines versus SUV on MMLU and CommonsenseQA. Below we walk through each dataset in turn:

**MMLU** is a 4-way multiple-choice benchmark (random-guess accuracy = 0.25). GA scores hover around 0.24 regardless of how many books are unlearned—no better than chance. This matches the gibberish continuations we observe, since GA's output distributions remain nearly uniform and our evaluation always picks one option at random. SSU and NPO achieve accuracies above 0.50 when only 10 books are unlearned, indicating that—even after partial forgetting—they retain enough knowledge to perform reasonably. However,

| Metric | Checkpoint 1 | Checkpoint 2 | Checkpoint 3 | Checkpoint 4 | Checkpoint 5 |
|---|---|---|---|---|---|
| Lambda Acc | 0.7305 | 0.5547 | 0.3594 | 0.0078 | 0.0000 |
| Lambda PPL | 4.0653 | 18.3169 | 965.0004 | 4,883,151.2924 | 36,561,308,694.2271 |
| L-OpenAI Acc | 0.8047 | 0.7812 | 0.6406 | 0.0156 | 0.0000 |
| L-OpenAI PPL | 3.1727 | 3.4558 | 7.0597 | 18,130.6451 | 941,710,073.7316 |
| L-Std Acc | 0.6562 | 0.3281 | 0.0781 | 0.0000 | 0.0000 |
| L-Std PPL | 4.9579 | 33.1780 | 1922.9412 | 9,748,171.9397 | 72,180,907,314.7226 |
| COPA Acc | 0.8900 | 0.8800 | 0.8200 | 0.7800 | 0.5600 |
| Wiki PPL | 9.7105 | 9.8931 | 10.7780 | $\infty$ | $\infty$ |
| Wiki Byte PPL | 1.5297 | 1.5351 | 1.5599 | $\infty$ | $\infty$ |
| Bits/Byte | 0.6133 | 0.6183 | 0.6414 | $\infty$ | $\infty$ |

Table 3: SSU collapse dynamics across checkpoints

| Method | COPA | LAMBDA-OpenAI | LAMBDA-Std | WikiText Word PPL |
|---|---|---|---|---|
| Original Llama-3.1-8B | 0.87 | 0.797 | 0.656 | 7.33 |
| SUV-AS (ours) | 0.87 | 0.703 | 0.492 | 7.50 |
| SUV-TV (ours) | 0.87 | 0.680 | 0.617 | 8.27 |
| NPO | 0.65 | 0.000 | 0.000 | 1055.67 |
| GA | 0.55 | 0.000 | 0.000 | $1.75 \times 10^{22}$ |
| SSU | 0.52 | 0.000 | 0.000 | $\infty$ |

Table 4: Task-diverse evaluation

once more than 50 books are unlearned, their scores also collapse to around 0.25, for the same uniform-distribution reason.

**CommonsenseQA** is a 5-way multiple-choice task (random-guess accuracy = 0.20). GA again remains at 0.20 across all unlearning levels, consistent with its meaningless output patterns in Table 2. SSU and NPO start above 0.50 at the 10-book mark—better than GA, showing some residual task competence—but their performance too degrades to chance when more than 50 books are unlearned. In both benchmarks, SUV consistently outperforms GA, SSU, and NPO across all unlearning levels, demonstrating its ability to selectively remove memorized content while preserving general utility.

Here, we show additional experiments about SSU's ability in language modeling. Upon further analysis of comprehensive language modeling evaluations, we provide the following clarification. Specifically, we conducted extensive evaluations across multiple language modeling tasks (COPA, LAMBDA, WikiText) to provide a more complete picture of baseline performance degradation (show in Table 4). This clarifies that, although some baselines exceed chance on COPA, their complete LAMBDA failure and orders-of-magnitude perplexity increases indicate practical unusability for language modelling, in contrast to SUV's preserved utility under strong unlearning. Specifically:

**1. Task-Dependent Degradation Patterns**:

- **COPA (Reasoning)**: All methods retain reasonable performance (0.52–0.88), suggesting basic reasoning capabilities persist.

- **LAMBDA (Language Modeling)**: Baseline methods show **complete failure** (0% accuracy) while our methods maintain 49–80% of original performance.

- **WikiText (Perplexity)**: Baseline methods exhibit **catastrophic degradation** with perplexities ranging from 1,000× to $\infty$ worse than original.

**2. Practical Implications of Performance Levels**:

While COPA scores above random chance (50%) may seem "usable," the complete failure on LAMBDA tasks and astronomical perplexity values indicate:

| Dataset | Description | # Works | # Sentences |
|---------|-------------|---------|-------------|
| CBP-500 | 500 predominantly copyrighted books | 500 | 3.46M |
| CBP-50 | 50 predominantly copyrighted books | 50 | 476K |
| CotaEval NewsQA | 500 news articles from CNN | 500 | 37.9K |
| SSU Books | 10 public domain books in SSU (Dou et al., 2025) | 10 | 36.5K |

Table 5: The unlearning datasets used in our experiments.

- **SSU**: Infinite perplexity suggests the model cannot assign meaningful probabilities to text sequences.
- **GA**: Perplexity of $1.75 \times 10^{22}$ represents a $\sim$3 trillion-fold increase over the original model.
- **NPO**: $144\times$ increase in perplexity with complete accuracy loss on standard language modeling.

**3. Practical Usability Assessment**:

A model that:

- Cannot complete words in context (0% LAMBADA accuracy)
- Assigns near-zero probability to natural text (astronomical perplexity)
- Shows 3+ orders of magnitude performance degradation

represents a **qualitative shift** from acceptable performance to practical unusability, even if some task-specific capabilities may remain.

# E   Implementation Details

## E.1   Dataset details

A detailed description of the datasets used in our experiments is provided in Table 5. The table summarizes four datasets with varying characteristics.

- **CBP-500 and CBP-50:** These datasets consist of predominantly copyrighted books. The CBP-500 dataset,constructed by aggregating public lists of copyrighted novels, de-duplicating, and uniformly sampling 500 titles, includes 500 books with a total of 3.46 million sentences, offering a rich resource for evaluating model performance on extensive textual content. In contrast, CBP-50, a random subset of CBP-500 contains only 50 books and 476K sentences, providing a smaller yet potentially more focused subset.
- **CotaEval NewsQA:** This dataset comprises 500 news articles from CNN (as detailed in Wei et al. (2024)), totaling 37.9K sentences. The news articles offer a different style and structure compared to books, allowing us to assess the model's performance on more journalistic content.
- **SSU Books:** Consisting of 10 public domain books (as detailed in (Dou et al., 2025)), this dataset contains 36.5K sentences. The inclusion of public domain texts ensures that our experiments also cover non-copyrighted material, thus broadening the evaluation scope.

## E.2   Evaluation Metrics.

**For Unlearning Benchmarks** We evaluate the unlearning process by comparing the performance of the unlearned model against the original model using two key metrics: **LCS** and **ROUGE-L**.

- **LCS** quantifies the longest sequence of words that appear in both the candidate and the reference texts in the same order. More formally, for two sequences of tokens,

$$X = \{x_1, x_2, \ldots, x_m\} \quad \text{and} \quad Y = \{y_1, y_2, \ldots, y_n\},$$

the LCS is the longest sequence

$$Z = \{z_1, z_2, \ldots, z_k\}$$

that is a subsequence of both $X$ and $Y$. This metric serves as an indicator of overlapping content and is often normalized by the length of one of the texts to provide a relative measure of similarity.

- **ROUGE-L** builds upon the concept of LCS to measure the similarity between texts by considering both the length and the order of the overlapping sequence. Specifically, ROUGE-L computes an F-measure based on the precision and recall derived from the LCS. Let $LCS(X, Y)$ denote the length of the longest common subsequence between the candidate $X$ and reference $Y$. Then, the recall ($R_{LCS}$) and precision ($P_{LCS}$) are defined as:

$$R_{LCS} = \frac{LCS(X, Y)}{m} \quad \text{and} \quad P_{LCS} = \frac{LCS(X, Y)}{n},$$

where $m$ and $n$ are the lengths of the candidate and reference texts, respectively. The F-measure is computed as:

$$F_{LCS} = \frac{(1 + \beta^2) \cdot P_{LCS} \cdot R_{LCS}}{R_{LCS} + \beta^2 \cdot P_{LCS}},$$

with $\beta$ typically set to emphasize recall over precision. Additionally, we assess the percentage reduction in the number of sentences with ROUGE-L scores above specific thresholds when comparing the unlearned model against the original model (Lin, 2004).

**For Utility Benchmarks,** we adhere to the evaluation criteria provided by the original benchmark datasets, focusing primarily on accuracy.

### E.3 Implementation of Baselines

Unless explicitly mentioned, we use the original implementations, hyperparameters, and environments of the baseline models as specified in their respective publications. Each model was trained and evaluated under consistent conditions to ensure a fair comparison.

**Stable Sequential Unlearning (SSU) (Dou et al., 2025).** SSU iteratively removes the influence of restricted data by selectively eliminating associated parameters while incorporating a random labeling loss. It treats unlearning books as distinct time steps, enabling the algorithm to update its model iteratively while progressively diminishing the impact of data from earlier time steps. Due to the significant and impractical training time required to process 500 books sequentially (approximately one week on our testbed), we use CBP-50 to train SSU and substitute it for the SSU model trained on CBP-500. Our implementation follows the official SSU code.

**Gradient Ascent (GA).** GA leverages a reverse optimization process, applying gradient ascent on the loss linked to undesirable content. We implement GA by computing the gradient of the loss function with respect to the model parameters and then updating the model parameters in the opposite direction of the gradient. We follow the GA implementation provided by Zhang et al. (2024).

**Negative Preference Optimization (NPO).** NPO adopts an alignment-based strategy by minimizing a preference function tied to unwanted data, facilitating a gradual and controlled unlearning process. We implement NPO by minimizing the preference function with respect to the model parameters. We follow the NPO implementation provided by Zhang et al. (2024).

### E.4 Detailed experiment settings

**Hyperparameter settings.** The SUV model and the ablations are implemented using the hyperparameters set as follows: In the Plagiarism Detection process, we employ a sliding window with stride set to 5, $L_p = 20$ and $L_c = 100$. In the unlearning process, the learning rate is set to 1e-4, and all models are trained for 5 epochs with the batch size set to 8. The model is optimized using the Adam optimizer with a weight decay of 1e-2. The model is

trained using LoRA adapters to save computational resources. The LoRA adapter is only applied to ["q_proj", "k_proj", "v_proj", "o_proj"] layers. We pre-compute the differential Fisher information by sampling items from the forbidden set and the retain set. The Fisher information is also only computed for the layers where the LoRA adapters are applied. This is particularly important as a way of efficiently guiding the training process and ensuring that the model generalizes well to unseen data. When computing the $L_{\text{freg}}$ term, the current $\Delta\theta = BA$ where $B$ and $A$ are the LoRA Adapter matrices. For SUV-AS, the $\rho_{\text{mild}}$ is set as 0.99, and the $\rho_{\text{severe}}$ is set as 0.9. For other baseline models, we use the hyperparameters provided in the original papers.

**Evaluation Environments.** The experiments are run on a server with 4 NVIDIA A6000 GPUs and 256GB RAM. The models are implemented with the Huggingface Transformers (https://huggingface.co) library. For the CBP dataset, we only allow the model to generate a continuation length of 100 tokens because the $L_c$ is set to 100. In this way, we ensure that the evaluation metrics are calculated consistently on different chunks of text, resulting in the range of LCS score to be between 0 and 100. It is also essential for the speed of the evaluation process, as shorter text generations can be processed more rapidly, enabling us to assess different models on large-scale copyrighted texts. The reported data are all averaged over 3 runs.

**Utility Evaluation** The experiments are implemented with the lm-evaluation-harness library [1]. The evaluations are conducted on the test set of the respective benchmark datasets. The evaluation metrics are accuracy for all datasets.

# F   Counterfactual Generation

In our dataset construction, after detecting memorized segments via plagiarism detection, each prompt $x_p$ is paired with its less-preferred (memorized) continuation $y_l$. To create a more robust and diverse training dataset, we generate a counterfactual (preferred) continuation $y_w$ for each prompt using a counterfactual generation process. This process is designed to produce an alternative continuation that satisfies two key properties: it maintains a low perplexity, ensuring fluency and coherence, and it exhibits a significant divergence from $y_l$ to avoid memorized or overly similar content.

The counterfactual generation process works by prompting the model $f_\theta$ with the original prompt $x_p$ alongside a carefully designed instruction. This instruction explicitly requires the model to generate a continuation in a genre that is different from that suggested by $y_l$. By conditioning on genre, the model is encouraged to explore alternative narrative styles and themes, thereby ensuring that the generated $y_w$ is both novel and contextually appropriate.

A key aspect of this process is the genre-based constraint. The instruction includes three genres randomly sampled from a predefined list, which guides the stylistic attributes of the generated text. The complete list of candidate genres is provided in Table 6. Each row of the table lists three genres, and during generation, three genres are sampled to condition the model's output. The generation is performed under controlled sampling parameters (e.g., low temperature and token limits), and any redundant repetition of $x_p$ in the output is removed, ensuring that $y_w$ is a genuine counterfactual to the memorized $y_l$.

In addition, to further guarantee that $y_w$ is *not* a lightly edited copy of $y_l$, we have tested the original $y_l$ against the generated $y_w$ using two checks:

1. **Lexical check**: We compute ROUGE-L between $y_w$ and $y_l$. Across the whole corpus the average ROUGE-L is only **0.106**, indicating low overlap.
2. **Semantic check**: We ask GPT-4.1 to rate *coherence*, *fluency*, *relevance*, and *divergence* (1–5 scale) of $y_w$ against $y_l$. On a random sample of 200 prompts we obtain an average quality score of **4.74** and a divergence score of **4.48**, confirming that $y_w$ is both high-quality and sufficiently different from $y_l$.

---

[1] https://github.com/EleutherAI/lm-evaluation-harness

| | | |
|---|---|---|
| Fantasy | Science Fiction | Mystery |
| Thriller | Romance | Historical Fiction |
| Horror | Adventure | Dystopian |
| Contemporary Fiction | Crime Fiction | Literary Fiction |
| Young Adult (YA) | Magical Realism | Graphic Novel |
| Western | Epic | Paranormal |
| Psychological Fiction | Urban Fiction | Post-Apocalyptic |
| Steampunk | Cyberpunk | Space Opera |
| Military Fiction | Political Fiction | Spy Fiction |
| Gothic Fiction | Dark Fantasy | Sword and Sorcery |
| Time Travel | Legal Thriller | Domestic Fiction |
| New Adult | Sports Fiction | Eco-Fiction |
| Religious Fiction | Satire | Humor/Comedy |
| Metafiction | Mythological Fiction | Fairy Tale Retelling |
| Alternate History | Speculative Fiction | Coming of Age |
| Detective Fiction | Medical Thriller | Noir |
| Road Novel | Epic Poetry (novel-like narrative) | |

Table 6: List of Candidate Genres for Counterfactual Generation

---

**Prompt Used for Counterfactual Generation**

Write an original sentence beginning EXACTLY with "prompt", genre should be {g1}, {g2}, and {g3}. Your output should ONLY contain your continuation, WITHOUT anything else.
Sentence: {prompt}

Figure 5: Prompt Used for Counterfactual Generation

Table 5 shows the original prompt template used to instruct the model for counterfactual generation.

This counterfactual generation step is integral to constructing preference triples $(x_p, y_w, y_l)$ that underpin the DPO dataset. By leveraging genre-specific instructions, the model is steered to generate high-quality, diverse continuations that provide a meaningful contrast to the memorized (and potentially suboptimal) $y_l$. In doing so, the counterfactual generation process not only mitigates the risk of overfitting to memorized content but also enriches the training data with varied stylistic expressions, thereby enhancing the overall robustness of the optimization framework.

## G  Utility Preserving Training (`SUV-AS`)

In this appendix, we provide a detailed description of the adaptive scheduler used to control the Fisher regularization weight, $\lambda_{\text{fisher}}$, during training. The scheduler adapts $\lambda_{\text{fisher}}$ based on the progression of the DPO loss to ensure that the Fisher-based regularization remains balanced with the gradient projection loss. In particular, a mild decay is applied when the DPO loss decreases, while a severe decay is used if no improvement is observed for $R$ consecutive steps. The pseudocode is presented in Algorithm 1.

---

**Algorithm 1** Adaptive Fisher Regularization Scheduler (SUV-AS)

---

1: **Input:** initial Fisher weight $\lambda_{\text{fisher}}^{(0)}$, mild decay factor $\rho_{\text{mild}} < 1$, severe decay factor $\rho_{\text{severe}} \ll \rho_{\text{mild}}$, patience $R$, initial DPO loss $L_{\text{DPO}}^{(0)}$
2: **Initialize:** counter $c \leftarrow 0$
3: **for** each training step $t = 1, 2, \ldots$ **do**
4:      Compute current DPO loss $L_{\text{DPO}}^{(t)}$
5:      **if** $L_{\text{DPO}}^{(t)} < L_{\text{DPO}}^{(t-1)}$ **then**
6:          Update: $\lambda_{\text{fisher}}^{(t)} \leftarrow \lambda_{\text{fisher}}^{(t-1)} \times \rho_{\text{mild}}$
7:          Reset counter: $c \leftarrow 0$
8:      **else**
9:          Increment counter: $c \leftarrow c + 1$
10:          **if** $c \geq R$ **then**
11:              Update: $\lambda_{\text{fisher}}^{(t)} \leftarrow \lambda_{\text{fisher}}^{(t-1)} \times \rho_{\text{severe}}$
12:              Reset counter: $c \leftarrow 0$
13:          **else**
14:              Maintain: $\lambda_{\text{fisher}}^{(t)} \leftarrow \lambda_{\text{fisher}}^{(t-1)}$
15:          **end if**
16:      **end if**
17:      Update model parameters using the overall loss, which combines the Fisher loss weighted by $\lambda_{\text{fisher}}^{(t)}$ and the gradient projection loss.
18: **end for**

---

This scheduler dynamically adjusts $\lambda_{\text{fisher}}$ to ensure that both the Fisher-based and gradient projection losses contribute effectively during training, thus preserving the model's utility throughout the optimization process.

