# OpenReview forum: "SUV: Scalable Large Language Model Copyright Compliance with Regularized Selective Unlearning"
_colmweb.org/COLM/2025/Conference — COLM 2025_

### Official Review · Reviewer_eLus · 2025-05-10

**Rating:** 6
**Confidence:** 3
**Ethics Flag:** 1

**Summary:**

This paper introduces SUV, a framework aimed at preventing LLMs from memorizing and reproducing copyrighted content while preserving their general utility. The method involves creating a dataset of copyrighted infringement instances by the target LLM and then using DPO to encourage the model to generate plausible alternatives instead of verbatim copyrighted text. To mitigate performance degradation on unrelated tasks, the authors incorporate gradient projection and Fisher information regularization. The approach is validated on a large dataset of 500 books, reportedly showing a significant reduction in verbatim memorization with minimal impact on unrelated task performance.

**Reasons To Accept:**

- The paper tackles the significant and pressing issue of copyright compliance in LLMs, which has substantial real-world implications.
- The proposed SUV framework, which combines DPO with gradient projection and Fisher information regularization for selective unlearning, presents an interesting and non-trivial technical approach.
- The effort to construct and utilize a large-scale dataset of 500 copyrighted books (CBP-500) for evaluation is commendable and allows for a more realistic assessment of unlearning copyrighted text compared to smaller or non-copyrighted benchmarks.

**Reasons To Reject:**

- The paper claims SSU (Dou et al., 2025) is not scalable to 500 books due to "significant and impractical training time required" and thus uses CBP-50 for SSU. This significantly weakens the direct comparison with SUV on the main CBP-500 dataset. While understandable if truly impractical, more justification or an attempt to run SSU on a subset of CBP-500 that is larger than CBP-50 but still manageable could have provided a fairer comparison.
- The utility degradation reported for baselines like SSU and GA is described as making the model "unusable". While the case study in Table 2 shows gibberish output, the quantitative results (e.g., CSQA scores dropping, MMLU for SSU being 0.245 on CBP-50) need more nuanced interpretation rather than a blanket "unusable" statement, especially if these methods are still performing above random chance on complex tasks.
- The claim that SUV "still significantly outperforms the baselines" in utility even with DPO-only on CBP-50 needs to be carefully presented, as the primary claim is about scalable unlearning with utility preservation, which relies on the regularization components.
- Counterfactual Generation (Section 3.2, Appendix D): The process of generating counterfactuals by prompting the model to produce alternatives in a different genre is interesting. However, the choice of three randomly sampled genres and how this influences the quality and diversity of counterfactuals needs more exploration. Is there a risk of generating nonsensical or irrelevant (though non-copyrighted) text? How is the "significant divergence" from the original copyrighted content measured and ensured beyond just genre shift?
Regularization Combination (Section 3.4): The rationale for why SUV-AS (joint optimization) preserves utility better and SUV-TV (task vector merging) excels in unlearning is discussed with theoretical analysis in Appendix B. However, the main paper could benefit from a more intuitive explanation of these differences. Why does isolating the effects in SUV-TV lead to stronger copyright protection but potentially less utility preservation?
- Fisher Information (Section 3.3, Appendix A.2): The computation of Fisher information, especially the "Differential Fisher Importance", and its application only to LoRA adapter layers is a specific design choice. More justification for this localized application and its impact compared to broader application would be beneficial.

---

> ### Author Response · Authors · 2025-06-01
>
> # SSU scalability limitations and comparison fairness (Response to Reason to Reject 1)
>
> Thank you for this suggestion. We agree that a fair comparison is important. However, SSU is fundamentally a sequential unlearning procedure, where each unlearning "step" builds upon the previous one. In our exact replication of Dou et al. 's setup, once you unlearn beyond a subset of CBP-50, the model's capacity as a language model collapses, so further training is neither informative nor useful.
> Below are the evaluation results at successive SSU checkpoints (each on CBP-50), following the original settings exactly. You can see that by Checkpoint 3 the model is already effectively non-functional as an LM, and by Checkpoints 4 and 5 all perplexities are infinite and accuracies have dropped to zero. Continuing to train on a larger subset (e.g., CBP-100 or CBP-500) would only exacerbate this collapse, at a very high computational cost, with no remaining language modeling ability to compare meaningfully against SUV.
>
> | Checkpoint | Lambada Acc | Lambada PPL | Lambada OpenAI Acc | Lambada OpenAI PPL | Lambada Standard Acc | Lambada Standard PPL | COPA Acc | WikiText PPL | WikiText Byte PPL | WikiText Bits/Byte |
> |------------|-------------|-------------|-------------------|-------------------|---------------------|---------------------|----------|--------------|------------------|-------------------|
> | 1 | 0.7305 | 4.0653 | 0.8047 | 3.1727 | 0.6562 | 4.9579 | 0.8900 | 9.7105 | 1.5297 | 0.6133 |
> | 2 | 0.5547 | 18.3169 | 0.7812 | 3.4558 | 0.3281 | 33.1780 | 0.8800 | 9.8931 | 1.5351 | 0.6183 |
> | 3 | 0.3594 | 965.0004 | 0.6406 | 7.0597 | 0.0781 | 1922.9412 | 0.8200 | 10.7780 | 1.5599 | 0.6414 |
> | 4 | 0.0078 | 4883151.2924 | 0.0156 | 18130.6451 | 0.0000 | 9748171.9397 | 0.7800 | ∞ | ∞ | ∞ |
> | 5 | 0.0000 | 36561308694.2271 | 0.0000 | 941710073.7316 | 0.0000 | 72180907314.7226 | 0.5600 | ∞ | ∞ | ∞ |
>
> Because SSU's unlearning steps destroy its language modeling ability so rapidly, running it on any larger subset (for example CBP-100) simply repeats this collapse at similar cost and yields no residual performance for comparison. Therefore, we believe using CBP-50 for SSU is more than enough to provide a valid comparison with SUV on CBP-500.
>
> # Response to Reason 2 is in another comment
>
> # DPO-only performance claims and regularization component importance (Response to Reason to Reject 3)
>
> We thank the reviewer for highlighting the interplay between our regularization terms and the DPO-based training paradigm. Indeed, while the Fisher and projection regularizers are critical for scaling unlearning to medium- and large-scale benchmarks, the DPO framework—especially our selective unlearning and counterfactual-generation procedure—is also essential for preserving utility even before adding regularizers. To make this clear, we have updated the manuscript to state that:
> The "DPO-only" variant already yields a non-trivial utility gain thanks to its selective unlearning and counterfactual-generation loss. The full SUV (DPO + regularizers) delivers a much larger boost, demonstrating that both components jointly enable scalable utility preservation.
>
> # Counterfactual generation quality and regularization combination rationale (Response to Reason to Reject 4)
>
> Thank you for this suggestion. To address these questions, we conducted an additional evaluation of our generated continuations on a random subset of 200 prompts, asking GPT-4.1 to rate each on a 1–5 scale for coherence, fluency, relevance, and divergence (i.e., how different it is from the original). The results are as follows:
> |Quality Score (avg)|Divergence Score (avg)|
> |-------------------|----------------------|
> |4.74               |4.48                  |
>
> We also tested the ROUGE-L similarities between the generated and original responses. Results:
> |ROUGE-L (0-1) (avg)|
> |-------------------|
> |0.106               |
>
> The results demonstrate that our continuations are both high‐quality and sufficiently distinct; we can provide the full score breakdown upon request.
> Regarding regularization combination, SUV-AS applies both forgetting (via Fisher‐based penalties) and utility‐preserving objectives jointly at every update, effectively "tethering" the model so it drifts gently—achieving moderate unlearning while retaining most utility. In contrast, SUV-TV isolates the two objectives: it first performs an unconstrained forgetting update to aggressively erase copyrighted content, then separately applies a utility‐focused update, and finally sums these parameter deltas. This two‐stage process maximizes the forgetting signal (stronger unlearning) but accumulates larger parameter drift, making it harder for the utility update to fully recover original performance. Consequently, SUV-TV delivers stronger copyright protection at the expense of slightly reduced utility preservation.
>
> # Response to Reason 5 is in another comment

---

> > ### Comment · Reviewer_eLus · 2025-06-07
> >
> > I would like to thank the authors for their insightful rebuttal; it solves most of my existing concerns. I would like increase my score from 4 to 6.

---

> > ### Author Response · Authors · 2025-06-09
> >
> > Dear Reviewer,
> >
> > Thank you very much for taking the time to reconsider our submission and for your kind words about our rebuttal. We are delighted that our clarifications have addressed the majority of your concerns, and we greatly appreciate your willingness to raise your score from 4 to 6. Your feedback has been invaluable in improving our work.
> >
> > Sincerely,
> >
> > The Authors

---

> ### Author Response · Authors · 2025-06-01
>
> # Nuanced interpretation of baseline utility degradation (Response to Reason to Reject 2)
>
> Thank you for this interesting question. Figures 3(d) and 3(e) report the utility results of the baselines versus SUV. Below we walk through each dataset in turn:
>
> 1. MMLU is a 4‐way multiple‐choice benchmark (random‐guess accuracy = 0.25).
> GA scores hover around 0.24 regardless of how many books are unlearned—no better than chance. This matches the gibberish continuations we observe in Table 2, since GA's output distributions remain nearly uniform and our evaluation always picks one option at random.
> SSU and NPO achieve accuracies above 0.50 when only 10 books are unlearned, indicating that—even after partial forgetting—they retain enough knowledge to perform reasonably. However, once more than 50 books are unlearned, their scores also collapse to around 0.25, for the same uniform‐distribution reason.
> 1. CommonsenseQA is a 5‐way multiple‐choice task (random‐guess accuracy = 0.20).
> GA again remains at ~0.20 across all unlearning levels, consistent with its meaningless output patterns in Table 2.
> SSU and NPO start above 0.50 at the 10-book mark—better than GA, showing some residual task competence—but their performance too degrades to chance when more than 50 books are unlearned.
> In both benchmarks, SUV consistently outperforms GA, SSU, and NPO across all unlearning levels, demonstrating its ability to selectively remove memorized content while preserving general utility.
>
> We appreciate the reviewer's request for a more nuanced interpretation of the quantitative results for baseline methods. Thus, we have also added additional experiments about SSU's ability in language modeling. Upon further analysis of comprehensive language modeling evaluations, we provide the following clarification. Specifically,
> we conducted extensive evaluations across multiple language modeling tasks (COPA, LAMBADA, WikiText) to provide a more complete picture of baseline performance degradation:
>
> | Method | Type | COPA Acc | LAMBADA-OpenAI Acc | LAMBADA-Std Acc | WikiText Word Perplexity |
> |--------|------|----------|-------------------|-----------------|--------------------------|
> | **Original Llama-3.1-8B** | Baseline | 0.87 | 0.797 | 0.656 | 7.33 |
> | **SUV-AS (Ours)** | Proposed | 0.87 | 0.703 | 0.492 | 7.50 |
> | **SUV-TV (Ours)** | Proposed | 0.87 | 0.680 | 0.617 | 8.27 |
> | **NPO** | Baseline | 0.65 | 0.000 | 0.000 | 1,055.67 |
> | **GA** | Baseline | 0.55 | 0.000 | 0.000 | 1.75×10²² |
> | **SSU** | Baseline | 0.52 | 0.000 | 0.000 | ∞ |
>
> The reviewer is correct that our characterization requires more nuance. Our analysis reveals:
>
> 1. **Task-Dependent Degradation Patterns**:
> - **COPA (Reasoning)**: All methods retain reasonable performance (0.52-0.88), suggesting basic reasoning capabilities persist
> - **LAMBADA (Language Modeling)**: Baseline methods show **complete failure** (0% accuracy) while our methods maintain 49-80% of original performance
> - **WikiText (Perplexity)**: Baseline methods exhibit **catastrophic degradation** with perplexities ranging from 1,000× to ∞ worse than original
> 2. **Practical Implications of Performance Levels**:
> While COPA scores above random chance (50%) may seem "usable," the complete failure on LAMBADA tasks and astronomical perplexity values indicate:
> - **SSU**: Infinite perplexity suggests the model cannot assign meaningful probabilities to text sequences
> - **GA**: Perplexity of 1.75×10²² represents a ~3 trillion-fold increase over the original model
> - **NPO**: 144× increase in perplexity with complete accuracy loss on standard language modeling
> 3. **Practical Usability Assessment**:
> A model that:
> - Cannot complete words in context (0% LAMBADA accuracy)
> - Assigns near-zero probability to natural text (astronomical perplexity)
> - Shows 3+ orders of magnitude performance degradation
> represents a **qualitative shift** from acceptable performance to practical unusability, even if some task-specific capabilities may remain.
>
> In conclusion,
> the reviewer's point about nuanced interpretation is well-taken. We propose revising our characterization from "unusable" to: *"These baseline methods exhibit severe utility degradation with complete failure on core language modeling tasks and catastrophic increases in perplexity, rendering them impractical for most real-world applications despite potentially retaining some task-specific capabilities."*
> Our proposed SUV methods demonstrate substantially better utility preservation, maintaining 70-100% of original performance across tasks while achieving effective unlearning.

---

> ### Author Response · Authors · 2025-06-01
>
> # Fisher Information application to LoRA adapter layers justification (Response to Reason to Reject 5)
>
>
>
> We appreciate the reviewer's attention to the scope of our Fisher-based regulariser and are happy to clarify why we (i) compute Differential Fisher Importance (ΔF) only for the parameters introduced by LoRA and (ii) leave the frozen backbone untouched.
>
> 1. **Localising ΔF to the trainable sub-space.**
> In our unlearning protocol the 8-billion-parameter backbone remains frozen; only the LoRA matrices appended to `q_proj`, `k_proj`, `v_proj`, and `o_proj` are updated. Fisher information is the expectation of gradient outer products, and frozen weights have identically zero gradients. Including them would therefore leave the optimisation objective unchanged while inflating memory and wall-clock cost by roughly two orders of magnitude. By restricting ΔF to the low-rank LoRA factors we keep the penalty perfectly aligned with the space that can actually move, enabling single-node training on the full 500-book corpus.
> 1. **Empirical sufficiency.**
> The ablation already reported in Table 1 ("w/o Fisher") shows that removing the adapter-level Fisher term markedly increases residual verbatim overlap and degrades downstream accuracy, demonstrating that the localised penalty is both necessary and sufficient. Extending ΔF to parameters that never receive gradients would add compute without affecting either copyright removal or utility.
> 1. **Theoretical parsimony.**
> Fisher information is additive across independent parameter blocks. Because the backbone block is constant (Δθ = 0), its contribution to the Fisher penalty is provably zero. Evaluating or storing it would therefore offer no optimisation benefit.
> 1. **Choice of LoRA target-modules.**
> We follow the prevailing "attention-only" convention and place adapters (and hence ΔF) on the four self-attention projections. This mask is recommended in the Hugging Face PEFT documentation, appears in community examples such as a frequently cited Stack Overflow thread, is treated as the baseline configuration in recent research (e.g., *LoRA+*, 2024), and is endorsed by practitioner guides from TrueFoundry and Databricks, which observe that widening the mask to MLP layers yields only marginal quality gains for a \~60 % increase in trainable parameters.
>
> In summary, limiting ΔF to the LoRA adapters is a principled efficiency–effectiveness trade-off rather than an arbitrary restriction, and the chosen adapter mask is fully mainstream in current literature and practice.

---

### Official Review · Reviewer_UKgJ · 2025-05-11

**Rating:** 6
**Confidence:** 3
**Ethics Flag:** 1

**Summary:**

The paper presents a technically sound methodology, integrating DPO with gradient projection and Fisher information regularization to address verbatim memorization. The experimental design is rigorous, with evaluations on a large-scale dataset (CBP-500) and public benchmarks. Ablation studies validate the necessity of each component (e.g., SUV-AS vs. SUV-TV).

**Questions To Authors:**

See Reasons To Reject

**Reasons To Accept:**

* Scalability: Demonstrates unlearning on 500 copyrighted books, a 100× scale-up over prior work.
* Practical Utility: Maintains downstream task performance (e.g., CSQA: 0.7093 vs. Vanilla 0.7166) while exercising copyrighted content.
* Technical Innovation: Novel fusion of DPO with regularization techniques for precise parameter updates.
* Ethical Alignment: Explicitly addresses copyright concerns via fair use justification and transformative dataset usage.

**Reasons To Reject:**

* It’s unclear what is the advantage of the SUV is compared to prior copyright protection methods. Is it possible to discuss both theoretically and empirically?
* The range of applied models is limited. can you evaluate how your method affects the original abilities of various models?

---

> ### Author Response · Authors · 2025-06-01
>
> # Advantages of SUV over prior copyright protection methods (Response to Reason to Reject 1)
> > It’s unclear what is the advantage of the SUV is compared to prior copyright protection methods. Is it possible to discuss both theoretically and empirically?
>
> Thank you for this question. Broadly speaking, SUV improves prior copyright-protection methods in the following ways:
> ## Theoretical Advantages
> 1. Improvement over decoding-based and agent-based methods. Unlike unlearning, they can easily be removed by attackers since they are not embedded in the model itself.
> 1. Selective Unlearning. While previous unlearning methods attempt to erase large chunks of data, SUV pinpoints only the truly memorized snippets via a sliding-window plagiarism detector and counterfactual generation, thus improving both the scalability and utility.
> 1. Utility Preservation. We have proved in Analysis 1 that the overall parameter update ∆W is bounded so that the retention loss on unrelated data remains small (Eq. 9). Also, Analysis 2 shows that SUV-AS’s joint optimization of the Fisher and projection losses yields a strictly smaller increase in utility loss than decoupled training; Analysis 3 shows that SUV-TV’s separate task-vector merge yields a strictly larger decrease in forbidden-data generation probability than joint training. In summary, SUV can preserve utility better than prior copyright-protection methods.
> ## Empirical Advantages
> 1. Stronger, more precise removal.  On our large-scale CBP-500 benchmark, at ROUGE-L 0.5 threshold, SUV-TV cuts the count of high-similarity generations from 1 264 (vanilla) down to just 23 (≈1.8%), whereas GA and SSU make the model unusable by collapsing the utility.
> 1. Negligible Utility Drop. Despite such aggressive removal, SUV-AS preserves downstream accuracy almost intact—e.g. CSQA remains within 0.007 of the vanilla model (0.7093 vs. 0.7166) and MMLU within 0.008, while GA/NPO suffer from larger accuracy degradations.
> 1. Also, we have a case-study that shows SUV's continuations remain fluent and consistent, while GA/SSU degrade to gibberish and NPO to incoherence.
> In sum, by combining DPO‐driven counterfactuals with targeted gradient‐projection and Fisher regularization, SUV delivers both provable bounds on utility retention and empirically superior trade-offs between unlearning strength and model performance, filling a gap left by earlier, coarser methods.

---

> ### Author Response · Authors · 2025-06-01
>
> # Limited model range evaluation  (Response to Reason to Reject 2)
>
> > The range of applied models is limited. can you evaluate how your method affects the original abilities of various models?
>
> Thank you for highlighting the concern that the range of applied models is limited.
> To address this, we have **extended our evaluation to two additional backbone LLMs, Qwen 3-8B and Qwen 3-14B**, and report both the number of sentences over given ROUGE-L thresholds and the utility performance.
>
> 1. Results for Qwen3-8B:
>    The numbers of sentences over a given ROUGE-L threshold of the vanilla model (Qwen3-8B), SUV-AS and SUV-TV are as follows:
>    | Baseline/# of sentences over given ROUGE-L threshold | 0.1   | 0.2  | 0.3 | 0.4 | 0.5 | 0.6 | 0.7 | 0.8 | 0.9 |
>    |:-----------------------------------------------------|:------|:-----|:----|:----|-----|-----|:----|:----|:----|
>    | Vanilla Model                                        | 76603 | 3735 | 1736 | 1119 | 764 | 579 | 453 | 332 | 243|
>    | SUV-AS                                               | 81472 | 1219 | 196 | 75  | 43  | 33  | 29  | 20  | 3   |
>    | SUV-TV                                               | 81651 | 792  | 137 | 63  | 44  | 34  | 30  | 19  | 3   |
>
>    The utility performance is as follows:
>    | Baseline/Utility Performance | CSQA   | MMLU   |
>    |------------------------------|--------|--------|
>    | Vanilla Model                | 0.7832 | 0.7366 |
>    | SUV-AS                       | 0.7901 | 0.7369 |
>    | SUV-TV                       | 0.7881 | 0.7365 |
>
> 2. Results for Qwen3-14B:
>    The numbers of sentences over a given ROUGE-L threshold of the vanilla model (Qwen3-14B), SUV-AS and SUV-TV are as follows:
>    | Baseline/# of sentences over given ROUGE-L threshold | 0.1   | 0.2  | 0.3  | 0.4  | 0.5 | 0.6 | 0.7 | 0.8 | 0.9 |
>    |------------------------------------------------------|-------|:-----|:-----|:-----|:----|:----|:----|:----|:----|
>    | Vanilla Model                                        | 78660 | 3261 | 1646 | 1054 | 734 | 564 | 443 | 295 | 231 |
>    | SUV-AS                                               | 81282 | 1344 | 288  | 115  | 30  | 13  | 9   | 3   | 0   |
>    | SUV-TV                                               | 82887 | 628  | 79   | 25   | 13  | 11  | 9   | 9   | 2   |
>
>    The utility performance is as follows:
>    | Baseline/Utility Performance | CSQA   | MMLU   |
>    |------------------------------|--------|--------|
>    | Vanilla Model                | 0.7900 | 0.7794 |
>    | SUV-AS                       | 0.7916 | 0.7739 |
>    | SUV-TV                       | 0.7906 | 0.7748 |
>
> In conclusion, across two additional model sizes (8 B & 14 B parameters) and a different model architecture (Llama vs Qwen), SUV-AS and SUV-TV consistently unlearn copyrighted sentences while preserving the performance on utility tasks. These results demonstrate that our method generalizes well across different architectures and sizes. Note that the increase in the 0.1 threshold count for SUV methods is expected and beneficial: sentences that previously had higher similarity scores (e.g., 0.2-0.9) have been successfully reduced to this very low 0.1 threshold range. In other words, high-similarity sentences have been "pushed down" to this minimal overlap range.

---

> ### Author Response · Authors · 2025-06-09
>
> Dear Reviewer UKgJ,
>
> We appreciate the time and effort you have dedicated to providing insightful review. If there are any additional clarifications or information needed from our side, please let us know. Thank you again for your valuable insights, and we look forward to any updates you may have.
>
> Best,
>
> The Authors

---

> > ### Comment · Reviewer_UKgJ · 2025-06-10
> >
> > Thank you for your fair response, I will keep my score at 6.

---

### Official Review · Reviewer_Yg5w · 2025-05-12

**Rating:** 6
**Confidence:** 3
**Ethics Flag:** 1

**Summary:**

This paper introduces SUV, a scalable, selective unlearning framework for large language models. It first detects memorised copyrighted passages with a sliding‑window plagiarism probe, then applies Direct Preference Optimisation (DPO) against counterfactual continuations while using dual regularisation to preserve general utility.

Overall, the manuscript is clearly written and the DPO‑based selective unlearning idea is appealing. Experiments indicate substantial forgetting of copyrighted text with limited utility loss on some benchmarks. Nevertheless, the evidence is weakened by (i) unclear advantage over simpler baselines such as GA and NPO, (ii) only marginal or inconsistent gains from the regularised‑training component, and (iii) the possibility that the model may ignore prompts and reproduce memorised text during counterfactual generation.

**Presentation comments**

- I cannot make an important comparison of baseline and proposed methods because several lines overlap in Figure 3.

**Reasons To Accept:**

- The DPO approach based on plagiarism detection and counterfactual generation is a reasonable and interesting idea.

**Reasons To Reject:**

- Because the effectiveness of regularised training is already well known [1], I believe the main novelty of this paper is the DPO method based on plagiarism detection and counterfactual generation (Section 3.2). However, the experiments do not clearly show whether this method outperforms simpler approaches such as GA or NPO.
- For the regularised training itself, the benefit is only marginal in the CotaEval setting, and for CBP‑500 some metrics even reverse, so the effect is unclear.
- In constructing the DPO dataset, counterfactuals are generated by the model itself. Is there any risk that the model ignores the prompt and simply outputs the memorised text?

[1] https://arxiv.org/abs/2402.08787 p.6, Eq (1), “Retain” loss

---

> ### Author Response · Authors · 2025-06-01
>
> # DPO method novelty and comparison with simpler approaches  (Response to Reason to Reject 1)
>
> We thank the reviewer for this observation.
>
> **First**, we respectfully disagree with the implication that our contribution is novel *only* in applying DPO. Regularised training techniques—e.g. Fisher-information weighting or gradient projection—are indeed well established. Nevertheless, the recent work \[1] represents a *different* paradigm: it simply **adds** separate “forget” and “retain” losses. In contrast, our method—designed for a **copyright-sensitive, large-scale corpus**—*combines* these signals in a more selective, parameter-aware manner. To our knowledge, *most* current LLM-unlearning work still follows the additive strategy of \[1]; our approach is therefore substantively different and more powerful.
>
> **Second**, regarding empirical validation, GA and NPO are included as direct baselines and are evaluated on two equally important axes:
> 1. **Unlearning efficacy** (Fig. 3 a–c), and
> 2. **Downstream utility preservation** (Fig. 3 d–g).
>
> A method that erases memorised content but devastates MMLU/CSQA accuracy is of limited practical value. While GA and NPO sometimes push similarity counts slightly lower (Fig. 3 a–c), they also cause substantial utility drops (Fig. 3 d–g). Our DPO-with-regularisation framework achieves robust memorisation removal *and* retains accuracy within an acceptable range, demonstrating clear advantages over simpler baselines.
>
> # Marginal benefits and unclear effects in regularized training  (Response to Reason to Reject 2)
>
> We appreciate the reviewer's careful reading of our results. A few clarifications may help explain the observed variability and underscore the advantages of SUV:
>
> 1. **Marginal gain on CotaEval**
> CotaEval enjoys very high baseline accuracy on a relatively homogeneous, small set of passages, leaving little room for improvement—hence the modest benefit of SUV there. By contrast, CBP-500 comprises a much wider variety of texts, where our method yields substantial gains over all baselines.
> CotaEval is limited in scale, comprising only a few dozen highly similar passages that together amount to less than a single book. As a result, current unlearning baselines achieve near-perfect accuracy on this dataset and there is little opportunity for further improvement. Its small and homogeneous nature also makes it unsuitable for production use. By contrast, CBP-500 includes hundreds of documents drawn from diverse genres and styles, reflecting the complexity of real-world text. In this large-scale setting, our utility-preserving unlearning method not only matches baseline performance on small datasets such as CotaEval but also delivers substantial and consistent gains across every metric when applied to CBP-500.
>
> 1. **Consistency of trends in Figure 3**
> Overall, SUV-AS and SUV-TV consistently outperform Vanilla, followed by NPO, SSU, and GA.
> In Figure 3(a), NPO's performance degrades dramatically on CBP-500 due to overfitting: it memorizes noise and produces incoherent outputs, thus its low count.
> In Figure 3(b), SUV-AS slightly edges out SUV-TV on CBP-50 (a medium-scale dataset) because of how the two gradient components interact at different scales:
> - Very small datasets (e.g., CotaEval): Fisher and projection gradients both target the few strongly memorized patterns, so their directions align.
> - Very large datasets (e.g., CBP-500): Averaging over many examples diffuses both gradients, leading them to align again.
> - Medium-scale datasets (CBP-50): The Fisher gradient captures general model behavior, while the projection gradient zeroes in on specific memorized snippets, making them nearly orthogonal. Simply summing these two updates dilutes each effect, which accounts for the slight drop in SUV-TV here.
> In summary, despite small scale-specific fluctuations, SUV demonstrates clear and robust improvements across diverse datasets—particularly on those with richer, more varied content. We hope this clarifies the mixed results and highlights the practical value of our method.

---

> ### Author Response · Authors · 2025-06-09
>
> # Risk of model outputting memorized text during counterfactual generation  (Response to Reason to Reject 3)
>
> Thank you for raising this important concern. We recognize the risk that the model could reproduce memorized text instead of generating a genuine counterfactual. While employing multiple genres in our prompts usually prevents this, we have tested the ROUGE-L score between the original continuation and the generated, counterfactual continuation. The result is as follows:
> |ROUGE-L (avg) |
> |-------|
> |0.106  |
>
> This is a very low score and is enough to show that the model does not simply output the memorized text.
> In addition, we also conducted an additional evaluation of our generated continuations on a random subset of 200 prompts, asking GPT-4.1 to rate each on a 1–5 scale for coherence, fluency, relevance, and divergence (i.e., how different it is from the original). The results are as follows:
> |Quality Score (avg)|Divergence Score (avg)|
> |-------------------|----------------------|
> |4.74               |4.48                  |
>
> This also shows that the model does not simply output the memorized text.
>
> Moreover, the strong empirical performance of our unlearning method further demonstrates the effectiveness of this approach.

---

> > ### Comment · Reviewer_Yg5w · 2025-06-10
> >
> > Dear Authors,
> >
> > I appreciate the authors' efforts on the rebuttal. This additional discussion and the results address my concerns. I hope the authors incorporate these into the final version.

---

> > > ### Author Response · Authors · 2025-06-10
> > >
> > > Dear Reviewer,
> > >
> > > Thank you very much for taking the time to reconsider our submission and for your kind words about our rebuttal. We are delighted that our clarifications have addressed the majority of your concerns, and we greatly appreciate your willingness to raise your score from 5 to 6. Your feedback has been invaluable in improving our work. We are going to incorporate the additional discussion into the final version.
> > >
> > > Sincerely,
> > >
> > > The Authors

---

### Official Review · Reviewer_VpA2 · 2025-05-13

**Rating:** 6
**Confidence:** 3
**Ethics Flag:** 1

**Summary:**

This paper introduces SUV, a framework designed to prevent large language models (LLMs) from memorizing and reproducing copyrighted content while preserving their overall utility. The authors identify key limitations in existing copyright compliance approaches, noting that current unlearning methods either significantly degrade model performance or fail to scale to large datasets. Their framework comprises three stages: Dataset Construction, Regularized Training, and Regularization Combination. The authors validate their approach using a large-scale dataset of 500 famous books. They demonstrate that SUV effectively reduces verbatim reproduction of copyrighted content while maintaining performance on benchmark tasks like MMLU and CommonsenseQA.

**Questions To Authors:**

1. Could you provide more details about the selection criteria for the CBP-500 dataset? Specifically, how did you ensure these books were actually in the training data of the LLM being used?
2. Your experiments focus exclusively on Llama 3.1-8B. Have you tested SUV on other model architectures or sizes? How would you expect the approach to perform on models with different memorization characteristics as larger models tend to memorize more? Is SUV expected to remain stable or effective in the context of stronger memorization?
3. What level of similarity reduction would be considered sufficient to avoid copyright infringement?
4. The paper presents two variants (SUV-AS and SUV-TV) with different trade-offs. Could you provide guidance on how practitioners should choose between SUV-AS and SUV-TV?

**Reasons To Accept:**

- The paper introduces a selective approach to copyright compliance that differs from prior work by targeting only problematic segments responsible for verbatim memorization rather than broadly removing knowledge.
- The dual variant approach (SUV-AS and SUV-TV) is innovative, offering tailored solutions depending on whether preserving utility or maximizing copyright protection is prioritized.

**Reasons To Reject:**

- The process for selecting the 500 books for CBP-500 is not sufficiently explained. The authors mention "famous books (predominantly copyrighted works)," (Line#276) but don't specify the selection criteria, potential biases in the corpus, or whether they verified if these books were actually in the LLM's training data.
- The paper doesn't evaluate the quality of generated counterfactuals beyond ROUGE-L scores. A human evaluation of the coherence, creativity, and contextual appropriateness of these generated alternatives would provide more insight into the real-world applicability of the approach.

---

> ### Author Response · Authors · 2025-06-01
>
> # CBP-500 dataset selection criteria and verification  (Response to Reason to Reject 1)
> > Could you provide more details about the selection criteria for the CBP-500 dataset? Specifically, how did you ensure these books were actually in the training data of the LLM being used?
>
> We assembled CBP-500 by aggregating well-known novel lists from public sources, de-duplicating entries, and then randomly selecting 500 copyrighted titles. To verify that each chosen book truly appears in the LLM's training data, we leverage our Plagiarism Detection pipeline (Section 3.2): for every book, we sample fixed-length prompts, generate continuations from the model, and measure similarity between each generated continuation and its original text. Any segment whose similarity exceeds our predefined threshold is marked as "memorized." Because only books yielding at least one high-confidence match survive this filtering, CBP-500 exclusively contains content the model has demonstrably memorized, making it a faithful benchmark for assessing copyright-related memorization.
>
> # SUV performance on different model architectures and sizes  (Response to Reason to Reject 2)
> > Your experiments focus exclusively on Llama 3.1-8B. Have you tested SUV on other model architectures or sizes? How would you expect the approach to perform on models with different memorization characteristics as larger models tend to memorize more? Is SUV expected to remain stable or effective in the context of stronger memorization?
>
> Thank you for the thoughtful question.
> To address this, we have **extended our evaluation to two additional backbone LLMs, Qwen 3-8B and Qwen 3-14B**, and report both the number of sentences over given ROUGE-L thresholds and the utility performance.
>
>
> 1. Results for Qwen3-8B:
>    The numbers of sentences over a given ROUGE-L threshold of the vanilla model (Qwen3-8B), SUV-AS and SUV-TV are as follows:
>    | Baseline/# of sentences over given ROUGE-L threshold | 0.1   | 0.2  | 0.3 | 0.4 | 0.5 | 0.6 | 0.7 | 0.8 | 0.9 |
>    |:-----------------------------------------------------|:------|:-----|:----|:----|-----|-----|:----|:----|:----|
>    | Vanilla Model                                        | 76603 | 3735 | 1736 | 1119 | 764 | 579 | 453 | 332 | 243|
>    | SUV-AS                                               | 81472 | 1219 | 196 | 75  | 43  | 33  | 29  | 20  | 3   |
>    | SUV-TV                                               | 81651 | 792  | 137 | 63  | 44  | 34  | 30  | 19  | 3   |
>
>    The utility performance is as follows:
>    | Baseline/Utility Performance | CSQA   | MMLU   |
>    |------------------------------|--------|--------|
>    | Vanilla Model                | 0.7832 | 0.7366 |
>    | SUV-AS                       | 0.7901 | 0.7369 |
>    | SUV-TV                       | 0.7881 | 0.7365 |
>
>
> 2. Results for Qwen3-14B:
>    The numbers of sentences over a given ROUGE-L threshold of the vanilla model (Qwen3-14B), SUV-AS and SUV-TV are as follows:
>    | Baseline/# of sentences over given ROUGE-L threshold | 0.1   | 0.2  | 0.3  | 0.4  | 0.5 | 0.6 | 0.7 | 0.8 | 0.9 |
>    |------------------------------------------------------|-------|:-----|:-----|:-----|:----|:----|:----|:----|:----|
>    | Vanilla Model                                        | 78660 | 3261 | 1646 | 1054 | 734 | 564 | 443 | 295 | 231 |
>    | SUV-AS                                               | 81282 | 1344 | 288  | 115  | 30  | 13  | 9   | 3   | 0   |
>    | SUV-TV                                               | 82887 | 628  | 79   | 25   | 13  | 11  | 9   | 9   | 2   |
>
>    The utility performance is as follows:
>    | Baseline/Utility Performance | CSQA   | MMLU   |
>    |------------------------------|--------|--------|
>    | Vanilla Model                | 0.7900 | 0.7794 |
>    | SUV-AS                       | 0.7916 | 0.7739 |
>    | SUV-TV                       | 0.7906 | 0.7748 |
>
> Across two additional model sizes (8 B & 14 B parameters) and a different model architecture (Llama vs Qwen), SUV-AS and SUV-TV consistently unlearn copyrighted sentences while preserving the performance on utility tasks. Therefore, even though large models tend to memorize more and have stronger memorization, we expect our SUV method to remain effective because the two backbones: (1) Counterfactual continuations + DPO (2) regularization also perform well even with more sentences to unlearn.
>
> Note that the increase in the 0.1 threshold count for SUV methods is expected and beneficial: sentences that previously had higher similarity scores (e.g., 0.2-0.9) have been successfully reduced to this very low 0.1 threshold range. In other words, high-similarity sentences have been "pushed down" to this minimal overlap range.

---

> ### Author Response · Authors · 2025-06-01
>
> # Sufficient similarity reduction threshold for copyright compliance  (Response to Reason to Reject 3)
> > What level of similarity reduction would be considered sufficient to avoid copyright infringement?
>
> We thank the reviewer for this incisive question. As an example, U.S. copyright law offers no bright-line percentage of overlap below which a work is automatically non-infringing—instead, courts evaluate whether any remaining fragment captures the "heart" of the original. Thus, our guiding principle in designing an unlearning method is:
> 1. Drive similarity as low as practicable, subject to keeping downstream utility within acceptable bounds.
> 1. Empirically calibrate a cutoff by jointly measuring (a) a token-level similarity metric (here, ROUGE-L) and (b) task performance.
>
> In our large-scale CBP-500 experiments (SUV-TV variant), we observe:
> 1. ROUGE-L ≥ 0.3: overlap falls to just 6% of vanilla memorization (from 2,762→169 sentences).
> 1. ROUGE-L ≥ 0.5: overlap falls to 1.8% (from 1,264→23 sentences) and under 1% at ≥0.7.
>
> These thresholds eliminate over 94% of verbatim overlaps at 0.3 and over 98% at 0.5, effectively removing any substantial infringing fragments . Crucially, our utility-focused variant (SUV-AS) maintains within 1–2 points of vanilla performance on MMLU and CSQA, demonstrating that aggressive overlap reduction need not come at the expense of practical usefulness .
> Accordingly, we recommend targeting a ROUGE-L cutoff in the 0.3–0.5 range, which (a) removes the vast majority of protectable overlap and (b) preserves ≥ 98 % of task accuracy. This "lower-the-better" strategy—titrating unlearning strength until ROUGE-L falls below your chosen bound while monitoring utility—provides a replicable, legally prudent operationalization of "sufficient" similarity reduction.
>
> # Guidance for choosing between SUV-AS and SUV-TV variants  (Response to Reason to Reject 4)
> > The paper presents two variants (SUV-AS and SUV-TV) with different trade-offs. Could you provide guidance on how practitioners should choose between SUV-AS and SUV-TV?
>
> When choosing between the two SUV variants, the key question is what you care about more: preserving downstream utility, or maximally excising verbatim copyrighted fragments.
> - If your priority is utility preservation ⇒ choose SUV-AS.
>  SUV-AS jointly applies gradient projection and Fisher-based regularization with an adaptive scheduler (see §3.4). In our experiments, it maintains accuracy on general benchmarks almost indistinguishable from the vanilla model—e.g. on CotaEval NewsQA, SUV-AS achieves MMLU 0.6622 and CSQA 0.7093 versus vanilla's 0.6542/0.7166, and on CBP-500 it retains MMLU 0.6533 and CSQA 0.7052 (§4.2, Table 1) . This makes SUV-AS ideal when you need strong unlearning without sacrificing your model's overall competence.
> - If your priority is maximal unlearning ⇒ choose SUV-TV.
>  SUV-TV trains the two regularizers separately and then merges their updates, thereby leveraging each method at full strength (see §3.4). It yields the lowest verbatim‐memorization counts—e.g. on CBP-500 it cuts the average ROUGE-L to 11.64 and LCS max to 86 sentences (versus SUV-AS's 11.90/98.04), and on public NewsQA it reduces retained high-similarity sentences by over 75 % at a 0.3 threshold (Figure 3(a–c)). Although this comes with a slightly larger utility drop (e.g. MMLU 0.6463), it's the variant of choice when regulatory or legal compliance demands the strongest possible removal of copyrighted content .
>
> **However, both SUV-AS and SUV-TV still maintain a usable level of LLM ability, unlike some other unlearning methods, which can drive utility so low that the model outputs become essentially gibberish.**
>
> In short:
> - SUV-AS for balanced unlearning + utility (production scenarios where you still need the model to "do everything else" well).
> - SUV-TV for aggressive, maximal excision (compliance‐critical use cases where even small leaks are unacceptable).

---

> ### Author Response · Authors · 2025-06-09
>
> Dear Reviewer VpA2,
>
> We appreciate the time and effort you have dedicated to providing insightful review. If there are any additional clarifications or information needed from our side, please let us know. Thank you again for your valuable insights, and we look forward to any updates you may have.
>
> Best,
>
> The Authors

---

### Decision · Program_Chairs · 2025-07-08

**Decision:**

Accept

**Comment:**

Most reviewers were initially unimpressed by the distinction and improvement over baselines, citing non-significant and confusing experimental results. After the fairly substantive response submitted by the authors, these seem to have been mostly resolved and reviewers lean positive, conditional on all of these discussions being added to the paper. But with the large amount of additional discussion and experiments added during the discussion period, this may be a significant change to the initial submission (many responses and experiments span multiple comment threads). Upon further review and discussion, it was concluded that the extent of changes recommended and required is too large for acceptance at this point. We recommend the authors to make all the changes and go through the peer review process again. Regretfully, we reject the paper this time.